# COptiDICE: Offline Constrained Reinforcement Learning via Stationary Distribution Correction Estimation

**Jongmin Lee**[1]*, **Cosmin Paduraru**[2], **Daniel J. Mankowitz**[2], **Nicolas Heess**[2], **Doina Precup**[2], **Kee-Eung Kim**[1], **Arthur Guez**[2]
[1]KAIST, [2]DeepMind

## Abstract

We consider the offline constrained reinforcement learning (RL) problem, in which the agent aims to compute a policy that maximizes expected return while satisfying given cost constraints, learning only from a pre-collected dataset. This problem setting is appealing in many real-world scenarios, where direct interaction with the environment is costly or risky, and where the resulting policy should comply with safety constraints. However, it is challenging to compute a policy that guarantees satisfying the cost constraints in the offline RL setting, since the off-policy evaluation inherently has an estimation error. In this paper, we present an offline constrained RL algorithm that optimizes the policy in the space of the stationary distribution. Our algorithm, COptiDICE, directly estimates the stationary distribution corrections of the optimal policy with respect to returns, while constraining the cost upper bound, with the goal of yielding a cost-conservative policy for actual constraint satisfaction. Experimental results show that COptiDICE attains better policies in terms of constraint satisfaction and return-maximization, outperforming baseline algorithms.

## 1 Introduction

Reinforcement learning (RL) has shown great promise in a wide range of domains, such as complex games (Mnih et al., 2015; Silver et al., 2017) and robotic control (Lillicrap et al., 2016; Haarnoja et al., 2018). However, the need to interact with the environment during learning hinders its widespread application to many real-world problems for which executing exploratory behavior in the environment is costly or dangerous. Offline RL (also known as batch RL) (Lange et al., 2012; Levine et al., 2020) algorithms sidestep this problem and perform policy optimization solely from a set of pre-collected data. The use of existing (offline) data can make offline reinforcement learning applicable to real world systems and and has led to a sharp increase in interest in this paradigm.

Recent works on offline RL, however, mostly assume that the environment is modeled as a Markov decision process (MDP), and standard offline RL algorithms focus on reward-maximization only (Fujimoto et al., 2019; Wu et al., 2019; Kumar et al., 2019; Siegel et al., 2020; Wang et al., 2020; Kumar et al., 2020). In contrast, in real-world domains it is common that the behavior of the agent is subject to additional constraints beyond the reward. Consider, for example, autonomous driving or an industrial robot in a factory. Some behaviors may damage the agent itself or its surroundings, and safety constraints should thus be considered as part of the objective of a suitable reinforcement learning system. One of the ways to mathematically characterize a constrained RL problem is through the formalism of constrained Markov Decision Processes (CMDP) (Altman, 1999). In CMDPs taking an action incurs a cost as well as a reward, and the goal is to maximize the expected long-term reward while satisfying a bound on the expected long-term cost. In this work, we aim to solve the constrained decision making problem in the *offline* RL setting, to enable deployment in various safety-critical domains where direct learning interactions are infeasible.

Offline constrained RL inherits the difficulties of offline unconstrained RL, while introducing additional challenges. First, since the target policy being optimized deviates from the data-collection

---

*Work done during an internship at DeepMind.

policy with no further data collection, distribution shift becomes the central difficulty. To mitigate the distributional shift, existing offline RL methods frequently adopt the pessimism principle: either by explicit policy (and critic) regularization that penalizes deviation from the data-collection policy (Jaques et al., 2019; Wu et al., 2019; Kumar et al., 2019; Siegel et al., 2020; Wang et al., 2020; Lee et al., 2020; Kostrikov et al., 2021) or by reward penalty to the uncertain state-action regions (Kumar et al., 2020; Kidambi et al., 2020; Yu et al., 2020). Second, in offline *constrained* RL, the computed policy should satisfy the given cost constraints when it is deployed to the real environment. Unfortunately off-policy policy evaluation inherently has estimation errors, and it is therefore difficult to ensure that a policy estimated from a finite dataset will satisfy the constraint when executed in the environment. In addition, constrained policy optimization usually involves an additional optimization for the Lagrange multiplier associated with the cost constraints. Actor-critic-based constrained RL algorithms thus have to solve triple (i.e. critic, actor, Lagrange multiplier) intertwined optimization problems (Borkar, 2005; Tessler et al., 2019), which can be very unstable in practice.

In this paper, we present an offline constrained RL algorithm that optimizes the state-action stationary distribution directly, rather than the Q-function or the policy. We show that such treatment obviates the need for multiple estimators for the value and the policy, yielding a single optimization objective that is practically solvable. Still, naively constraining the cost value may result in severe constraint violation in the real environment, as we demonstrate empirically. We thus propose a method to constrain the *upper bound* of the cost value, aiming to compute a policy more robust in constraint violation, where the upper bound is computed in a way motivated by a recent advance in off-policy confidence interval estimation (Dai et al., 2020). Our algorithm, *Offline Constrained Policy Optimization via stationary DIstribution Correction Estimation* (COptiDICE), estimates the stationary distribution corrections of the optimal policy that maximizes rewards while constraining the cost upper bound, with the goal of yielding a cost-conservative policy for better actual constraint satisfaction. COptiDICE computes the upper bound of the cost efficiently by solving an additional minimization problem. Experimental results show that COptiDICE attains a better policy in terms of constraint satisfaction and reward-maximization, outperforming several baseline algorithms.

## 2 BACKGROUND

A Constrained Markov Decision Process (CMDP) (Altman, 1999) is an extension of an MDP, formally defined by a tuple $M = \langle S, A, T, R, C = \{C_k\}_{1..K}, \hat{c} = \{\hat{c}_k\}_{1..K}, p_0, \gamma \rangle$, where $S$ is the set of states, $A$ is the set of actions, $T : S \times A \to \Delta(S)$ is a transition probability, $R : S \times A \to \mathbb{R}$ is the reward function, $C_k : S \times A \to \mathbb{R}$ is the $k$-th cost function with its corresponding threshold $\hat{c}_k \in \mathbb{R}$, $p_0 \in \Delta(S)$ is the initial state distribution, and $\gamma \in (0, 1]$ is the discount factor. A policy $\pi : S \to \Delta(A)$ is a mapping from state to distribution over actions. We will express solving CMDPs in terms of stationary distribution. For a given policy $\pi$, the stationary distribution $d^\pi$ is defined by:

$$d^\pi(s, a) = \begin{cases} (1 - \gamma) \sum_{t=0}^{\infty} \gamma^t \Pr(s_t = s, a_t = a) & \text{if } \gamma < 1, \\ \lim_{T \to \infty} \frac{1}{T+1} \sum_{t=0}^{T} \Pr(s_t = s, a_t = a) & \text{if } \gamma = 1, \end{cases} \quad (1)$$

where $s_0 \sim p_0$, $a_t \sim \pi(s_t)$, $s_{t+1} \sim T(s_t, a_t)$ for all timesteps $t \geq 0$. We define the value of the policy as $V_R(\pi) := \mathbb{E}_{(s,a) \sim d^\pi}[R(s, a)] \in \mathbb{R}$ and $V_C(\pi) := \mathbb{E}_{(s,a) \sim d^\pi}[C(s, a)] \in \mathbb{R}^K$. Constrained RL aims to learn an optimal policy that maximizes the reward while bounding the costs up to the thresholds by interactions with the environment:

$$\max_{\pi} V_R(\pi) \text{ s.t. } V_{C_k}(\pi) \leq \hat{c}_k \quad \forall k = 1, \dots, K \quad (2)$$

Lagrangian relaxation is typically employed to solve Eq. (2), leading to the unconstrained problem:

$$\min_{\lambda \geq 0} \max_{\pi} V_R(\pi) - \lambda^\top (V_C(\pi) - \hat{c}) \quad (3)$$

where $\lambda \in \mathbb{R}^K$ is the Lagrange multiplier and $\hat{c} \in \mathbb{R}^K$ is the vector-valued cost threshold. The inner maximization in Eq. (3) corresponds to computing an optimal policy that maximizes scalarized rewards $R(s, a) - \lambda^\top C(s, a)$, due to the linearity of the value function with respect to the reward function. The outer minimization corresponds to balancing the cost penalty in the scalarized reward

function: if the current policy is violating the $k$-th cost constraint, $\lambda_k$ increases so that the cost is penalized more in the scalarized reward, and vice versa.

In the offline RL setting, online interaction with the environment is not allowed, and the policy is optimized using the fixed offline dataset $D = \{(s_0, s, a, r, c, s')_i\}_{i=1}^N$ collected with one or more (unknown) data-collection policies. The empirical distribution of the dataset is denoted as $d^D$, and we will abuse the notation $d^D$ for $s \sim d^D$, $(s, a) \sim d^D$, $(s, a, s') \sim d^D$. We abuse $(s_0, s, a, s') \sim d^D$ for $s_0 \sim p_0, (s, a, s') \sim d^D$. We denote the space of data samples $(s_0, s, a, s')$ as $X$.

A naive way to solve (3) in an offline manner is to adopt an actor-critic based offline RL algorithm for $\max_\pi V_{R-\lambda^\top C}(\pi)$ while jointly optimizing $\lambda$. However, the intertwined training procedure of off-policy actor-critic algorithms often suffers from instability due to the compounding error incurred by bootstrapping out-of-distribution action values in an offline RL setting (Kumar et al., 2019). The instability would be exacerbated when the nested optimization for $\lambda$ is added.

## 3 OFFLINE CONSTRAINED RL VIA STATIONARY DISTRIBUTION CORRECTION ESTIMATION

In this section, we present our offline constrained RL algorithm, *Constrained Policy **Opti**mization via stationary **DI**stribution Correction Estimation* (COptiDICE). The derivation of our algorithm starts by augmenting the standard linear program for CMDP (Altman, 1999) with an additional $f$-divergence regularization:

$$\max_d \mathbb{E}_{(s,a)\sim d}[R(s,a)] - \alpha D_f(d||d^D) \tag{4}$$

$$\text{s.t. } \mathbb{E}_{(s,a)\sim d}[C_k(s,a)] \leq \hat{c}_k \qquad\qquad \forall k = 1, \ldots, K \tag{5}$$

$$\sum_{a'} d(s',a') = (1-\gamma)p_0(s') + \gamma \sum_{s,a} d(s,a)T(s'|s,a) \qquad\qquad \forall s' \tag{6}$$

$$d(s,a) \geq 0 \qquad\qquad \forall s, a, \tag{7}$$

where $D_f(d||d^D) := \mathbb{E}_{(s,a)\sim d^D}\left[f\left(\frac{d(s,a)}{d^D(s,a)}\right)\right]$ is the $f$-divergence between the distribution $d$ and the dataset distribution $d^D$, and $\alpha > 0$ is the hyperparameter that controls the degree of pessimism, i.e. how much we penalize the distribution shift, a commonly adopted principle for offline RL (Nachum et al., 2019b; Kidambi et al., 2020; Yu et al., 2020; Lee et al., 2021). We assume that $d^D > 0$ and $f$ is a strictly convex and continuously differentiable function with $f(1) = 0$. Note that when $\alpha = 0$, the optimization (4-7) reduces to the standard linear program for CMDPs. The Bellman-flow constraints (6-7) ensure that $d$ is the stationary distribution of a some policy, where $d(s,a)$ can be interpreted as a normalized discounted occupancy measure of $(s, a)$. Thus, we seek the stationary distribution of an optimal policy that maximizes the reward value (4) while bounding the cost values (5). Once the optimal solution $d^*$ has been estimated, its corresponding optimal policy is obtained by $\pi^*(a|s) = \frac{d^*(s,a)}{\sum_{a'} d^*(s,a')}$.

Now, consider the Lagrangian for the constrained optimization problem (4-7):

$$\min_{\lambda\geq 0,\nu} \max_{d\geq 0} \mathbb{E}_{(s,a)\sim d}[R(s,a)] - \alpha D_f(d||d^D) - \sum_{k=1}^K \lambda_k\left(\mathbb{E}_{(s,a)\sim d}[C_k(s,a)] - \hat{c}_k\right)$$

$$- \sum_{s'} \nu(s')\left[\sum_{a'} d(s',a') - (1-\gamma)p_0(s') - \gamma \sum_{s,a} d(s,a)T(s'|s,a)\right] \tag{8}$$

where $\lambda \in \mathbb{R}_+^K$ is the Lagrange multiplier for the cost constraints (5), and $\nu(s) \in \mathbb{R}$ is the Lagrange multiplier for the Bellman flow constraints (6). Solving (8) in its current form requires evaluation of $T(s'|s,a)$ for $(s,a) \sim d$, which is not accessible in the offline RL setting. To make the optimization tractable, we rearrange the terms so that the direct dependence on $d$ is eliminated, while introducing new optimization variables $w$ that represent stationary distribution corrections:

$$\min_{\lambda\geq 0,\nu} \max_{d\geq 0} \mathbb{E}_{\substack{(s,a)\sim d \\ s'\sim T(s,a)}} \left[R(s,a) - \lambda^\top C(s,a) + \gamma\nu(s') - \nu(s)\right] - \alpha\mathbb{E}_{(s,a)\sim d^D}\left[f\left(\frac{d(s,a)}{d^D(s,a)}\right)\right]$$

$$+ (1-\gamma)\mathbb{E}_{s_0\sim p_0}[\nu(s_0)] + \lambda^\top \hat{c}$$

$$= \min_{\lambda\geq 0,\nu} \max_{w\geq 0} \mathbb{E}_{(s,a)\sim d^D}\left[w(s,a)e_{\lambda,\nu}(s,a) - \alpha f(w(s,a))\right] + (1-\gamma)\mathbb{E}_{s_0\sim p_0}[\nu(s_0)] + \lambda^\top \hat{c} \tag{9}$$

where $e_{\lambda,\nu}(s,a) := R(s,a) - \lambda^\top C(s,a) + \gamma \mathbb{E}_{s' \sim T(s,a)}[\nu(s')] - \nu(s)$ is the advantage function by regarding $\nu$ as a state value function, and $w(s,a) := \frac{d(s,a)}{d^D(s,a)}$ is the stationary distribution correction. Every term in Eq. (9) can be estimated from samples in the offline dataset $D$:

$$\min_{\nu,\lambda \geq 0} \max_{w \geq 0} \mathbb{E}_{(s_0,s,a,s') \sim d^D} \left[ w(s,a)\hat{e}_{\lambda,\nu}(s,a,s') - \alpha f(w(s,a)) + (1-\gamma)\nu(s_0) \right] + \lambda^\top \hat{c} \quad (10)$$

where $\hat{e}_{\lambda,\nu}(s,a,s') := R(s,a) - \lambda^\top C(s,a) + \gamma\nu(s') - \nu(s)$ is the advantage estimate using a single sample. As a consequence, (10) can be optimized in a fully offline manner. Moreover, exploiting the strict convexity of $f$, we can further derive a closed-form solution for the inner maximization in (9) as follows. All the proofs can be found in Appendix B.

**Proposition 1.** *For any $\nu$ and $\lambda$, the closed-form solution of the inner maximization problem in* (9) *is given by:*

$$w^*_{\lambda,\nu}(s,a) = (f')^{-1} \left( \tfrac{1}{\alpha} e_{\lambda,\nu}(s,a) \right)_+ \ \text{where } x_+ = \max(0,x) \quad (11)$$

Finally, by plugging the closed-form solution (11) into (9), we obtain the following convex minimization problem:

$$\min_{\lambda \geq 0, \nu} L(\lambda,\nu) = \mathbb{E}_{(s,a) \sim d^D} \left[ w^*_{\lambda,\nu}(s,a)e_{\lambda,\nu}(s,a) - \alpha f(w^*_{\lambda,\nu}(s,a)) \right] + (1-\gamma)\mathbb{E}_{s_0 \sim p_0}[\nu(s_0)] + \lambda^\top \hat{c}$$

$$(12)$$

To sum up, by operating in the space of stationary distributions, constrained (offline) RL can in principle be solved by solving a single convex minimization (12) problem. This is in contrast to existing constrained RL algorithms that manipulate both Q-function and policy, and thus require solving triple optimization problems for the actor, the critic, and the cost Lagrange multiplier with three different objective functions. Note also that when $\lambda$ is fixed and treated as a constant, (12) reduces to OptiDICE (Lee et al., 2021) for unconstrained RL with the scalarized rewards $R(s,a) - \lambda^\top C(s,a)$, without considering the cost constraints. In order to meet the constraints, $\lambda$ should also be optimized, and the procedure of (12) can be understood as joint optimization of:

$$\nu \leftarrow \arg\min_\nu L(\lambda,\nu) \qquad\qquad \text{(OptiDICE for } R - \lambda^\top C) \quad (13)$$

$$\lambda \leftarrow \arg\min_{\lambda \geq 0} \lambda^\top \big( \hat{c} - \underbrace{\mathbb{E}_{(s,a) \sim d^D}[w^*_{\lambda,\nu}(s,a)C(s,a)]}_{\approx V_C(\pi)} \big) \qquad \text{(Cost Lagrange multiplier)}$$

Once the optimal solution of (12), $(\lambda^*, \nu^*)$, is computed, $w^*_{\lambda^*,\nu^*}(s,a) = \frac{d^{\pi^*}(s,a)}{d^D(s,a)}$ is also derived by (11), which is the stationary distribution correction between the stationary distribution of the optimal policy for the CMDP and the dataset distribution.

### 3.1 COST-CONSERVATIVE CONSTRAINED POLICY OPTIMIZATION

Our first method based on (12) relies on off-policy evaluation (OPE) using DICE to ensure cost constraint satisfaction, i.e. $\mathbb{E}_{(s,a) \sim d^D}[w_{\lambda,\nu}(s,a)C(s,a)] \approx \mathbb{E}_{(s,a) \sim d^\pi}[C(s,a)] \leq \hat{c}$. However, as we will see later, constraining the cost value estimate naively can result in constraint violation when deployed to the real environment. This is due to the fact that an off-policy value estimate based on a finite dataset inevitably has estimation error. Reward estimation error may be tolerated as long as the value estimates are useful as policy improvement signals: it may be sufficient to maintain the relative order of action values, while the absolute values matter less. For the cost value constraint, we instead rely on the estimated value directly.

To make a policy robust against cost constraint violation in an offline setting, we consider the constrained policy optimization scheme that exploits the *upper bound* of the cost value estimate:

$$\max_\pi \hat{V}_R(\pi) \ \text{ s.t. } \text{UpperBound}(\hat{V}_{C_k}(\pi)) \leq \hat{c}_k \ \ \forall k \quad (14)$$

Then, the key question is how to estimate the upper bound of the policy value. One natural way is to exploit bootstrap confidence interval (Efron & Tibshirani, 1993; Hanna et al., 2017). We can

construct bootstrap datasets $D_i$ by resampling from $D$ and run an OPE algorithm on each $D_i$, which yields population statistics for confidence interval estimation $\{\hat{V}_C(\pi)_i\}_{i=1}^m$. However, this procedure is computationally very expensive since it requires solving $m$ OPE tasks. Instead, we take a different approach in a more computationally efficient way motivated by CoinDICE (Dai et al., 2020), a recently proposed DICE-family algorithm for off-policy confidence interval estimation. Specifically, given that our method estimates the stationary distribution corrections $w(s, a) \approx \frac{d^\pi(s,a)}{d^D(s,a)}$ of the target policy $\pi$, we consider the following optimization problem for each cost function $C_k$:

$$\max_{\tilde{p} \in \Delta(X)} \mathbb{E}_{(s_0,s,a,s') \sim \tilde{p}}[w(s,a)C_k(s,a)] \tag{15}$$

$$\text{s.t. } D_{\text{KL}}(\tilde{p}(s_0,s,a,s')||d^D(s_0,s,a,s')) \leq \epsilon \tag{16}$$

$$\sum_{a'} \tilde{p}(s',a')w(s',a') = (1-\gamma)\tilde{p}_0(s') + \gamma \sum_{s,a} \tilde{p}(s,a)w(s,a)\tilde{p}(s'|s,a) \quad \forall s' \tag{17}$$

where $\tilde{p}(s_0, s, a, s') = \tilde{p}_0(s_0)\tilde{p}(s, a)\tilde{p}(s'|s, a)$ is the distribution over data samples $(s_0, s, a, s') \in X$ which lies in the simplex $\Delta(X)$, and $\epsilon > 0$ is the hyperparameter. In essence, we want to adversarially optimize the distribution over data samples $\tilde{p}$ so that it overestimates the cost value by (15). At the same time, we enforce that the distribution $\tilde{p}$ should not be perturbed too much from the empirical data distribution $d^D$ by the KL constraint (16). Lastly, the perturbation of distribution should be done in a way that maintains compatibility with the Bellman flow constraint. The constraint (17) is analogous to the Bellman flow constraint (6) by noting that $\tilde{p}(s,a)w(s,a) = \tilde{p}(s,a)\frac{d^\pi(s,a)}{d^D(s,a)} \approx d^\pi(s,a)$. In this optimization, when $\epsilon = 0$, the optimal solution is simply given by $\tilde{p}^* = d^D$, which yields the vanilla OPE result via DICE, i.e. $\mathbb{E}_{(s,a) \sim d^D}[w(s,a)C_k(s,a)]$. For $\epsilon > 0$, the cost value will be overestimated more as $\epsilon$ increases. Through a derivation similar to that for obtaining (12), we can simplify the constrained optimization into a single unconstrained minimization problem as follows. We denote $(s_0, s, a, s')$ as $x$ for notational brevity.

**Proposition 2.** *The constrained optimization problem (15-17) can be reduced to solving the following unconstrained minimization problem:*

$$\min_{\tau \geq 0, \chi} \ell_k(\tau, \chi; w) = \tau \log \mathbb{E}_{x \sim d^D}\left[\exp\left(\frac{1}{\tau}\big(w(s,a)(C_k(s,a) + \gamma\chi(s') - \chi(s)) + (1-\gamma)\chi(s_0)\big)\right)\right] + \tau\epsilon \tag{18}$$

*where $\tau \in \mathbb{R}_+$ corresponds to the Lagrange multiplier for the constraint (16), and $\chi(s) \in \mathbb{R}$ corresponds to the Lagrange multiplier for the constraint (17). In other words, $\min_{\tau \geq 0, \chi} \ell(\tau, \chi) = \mathbb{E}_{(s,a) \sim \tilde{p}^*}[w(s,a)C_k(s,a)]$ where $\tilde{p}^*$ is the optimal perturbed distribution of the problem (15-17). Also, for the optimal solution $(\tau^*, \chi^*)$, $\tilde{p}^*$ is given by:*

$$\tilde{p}^*(x) \propto d^D(x) \underbrace{\exp\left(\frac{1}{\tau^*}\big(w(s,a)(C_k(s,a) + \gamma\chi^*(s') - \chi^*(s)) + (1-\gamma)\chi^*(s_0)\big)\right)}_{=: \, \omega^*(x) \; (unnormalized \; weight \; for \; x = (s_0, s, a, s'))} \tag{19}$$

Note that every term in (18) can be estimated only using samples of the offline dataset $D$, thus it can be optimized in a fully offline manner. This procedure can be understood as computing the weights for each sample while adopting *reweighting* in the DICE-based OPE, i.e. $\text{UpperBound}(\hat{V}_C(\pi)) = \mathbb{E}_{x \sim d^D}[\tilde{\omega}^*(x)w(s,a)C(s,a)]$ where $\tilde{\omega}^*(x) = $ (normalized $\omega^*(x)$ of (19)). The weights are given non-uniformly so that the cost value is overestimated to the extent controlled by $\epsilon$.

**Remark.** *CoinDICE (Dai et al., 2020) solves the similar optimization problem to estimate an upper cost value of the target policy $\pi$ as follows:*

$$\max_{w \geq 0} \min_{\tau \geq 0, \nu} \tau \log \mathbb{E}_{\substack{x \sim d^D \\ a_0 \sim \pi(s_0) \\ a' \sim \pi(s')}}\left[\exp\left(\frac{1}{\tau}\big(w(s,a)(C_k(s,a) + \gamma\nu(s',a') - \nu(s',a')) + (1-\gamma)\nu(s_0,a_0)\big)\right)\right] + \tau\epsilon \tag{20}$$

*It is proven that (20) provides an asymptotic $(1-\alpha)$-upper-confidence-interval of the policy value if $\epsilon := \frac{\xi_\alpha}{N}$ where $\xi_\alpha$ is the $(1-\alpha)$-quantile of the $\chi^2$-distribution with 1 degree of freedom (Dai et al., 2020). Compared to our optimization problem (18), CoinDICE's (20) involves the additional outer maximization, which is for estimating $w(s,a) = \frac{\tilde{d}^\pi(s,a)}{\tilde{p}(s,a)}$, the stationary distribution corrections of the target policy $\pi$. In contrast, we consider the case when $w$ is given, thus solving the inner minimization alone is enough.*

Finally, we are ready to present our final algorithm COptiDICE, an offline constrained RL algorithm that maximizes rewards while bounding the *upper* cost value, with the goal of computing a policy robust against cost violation. COptiDICE addresses (14) by solving the following joint optimization.

$$\nu \leftarrow \arg\min_{\nu} L(\lambda, \nu) \qquad\qquad \text{(OptiDICE for } R - \lambda^\top C) \qquad (21)$$

$$\tau, \chi \leftarrow \arg\min_{\tau \geq 0, \chi} \sum_{k=1}^{K} \ell_k(\tau_k, \chi_k; w^*_{\lambda,\nu}) \qquad\qquad \text{(Upper cost value estimation)}$$

$$\lambda \leftarrow \arg\min_{\lambda \geq 0} \lambda^\top \big( \hat{c} - \underbrace{\ell(\tau, \chi; w^*_{\lambda,\nu})}_{\approx \text{UpperBound}(\hat{V}_C(\pi))} \big) \qquad\qquad \text{(Cost Lagrange multiplier)}$$

Compared to (13), the additional minimization for $(\tau, \chi)$ is introduced to estimate the upper bound of cost value.

## 3.2 POLICY EXTRACTION

Our algorithm estimates the stationary distribution corrections of the optimal policy, rather than directly obtaining the policy itself. Since the stationary distribution corrections do not provide a direct way to sample an action, we need to extract the optimal policy $\pi^*$ from $w^*(s,a) = \frac{d^{\pi^*}(s,a)}{d^D(s,a)}$, in order to select actions when deployed. For finite CMDPs, it is straightforward to obtain $\pi^*$ by $\pi^*(a|s) = \frac{d^{\pi^*}(s,a)}{\sum_{a'} d^{\pi^*}(s,a')} = \frac{d^D(s,a) w^*(s,a)}{\sum_{a'} d^D(s,a) w^*(s,a)}$. However, the same method cannot directly be applied to continuous CMDPs due to the intractability of computing the normalization constant. For continuous CMDPs, we instead extract the policy using importance-weighted behavioral cloning:

$$\max_{\pi} \mathbb{E}_{(s,a) \sim d^{\pi^*}}[\log \pi(a|s)] = \mathbb{E}_{(s,a) \sim d^D}[w^*(s,a) \log \pi(a|s)] \qquad (22)$$

which maximizes the log-likelihood of actions to be selected by the optimal policy $\pi^*$.

## 3.3 PRACTICAL ALGORITHM WITH FUNCTION APPROXIMATION

For continuous or large CMDPs, we represent our optimization variables using neural networks. The Lagrange multipliers $\nu$ and $\chi$ are networks parameterized by $\theta$ and $\phi$ respectively: $\nu_\theta : S \to \mathbb{R}$ is a feedforward neural network that takes a state as an input and outputs a scalar value, and $\chi_\phi : S \to \mathbb{R}^K$ is defined similarly. $\lambda \in \mathbb{R}_+^K$ and $\tau \in \mathbb{R}_+^K$ are represented by $K$-dimensional vectors. For the policy $\pi_\psi$, we use a mixture density network (Bishop, 1994) where the parameters of a Gaussian mixture model are output by the neural network. The parameters of the $\nu_\theta$ network are trained by minimizing the loss:

$$\min_{\theta} J_\nu(\theta) = \mathbb{E}_{x \sim d^D}\big[\hat{w}(s,a,s')(R(s,a) - \lambda^\top C(s,a) + \gamma \nu_\theta(s') - \nu_\theta(s)) \qquad (23)$$

$$- \alpha f(\hat{w}(s,a,s')) + (1-\gamma)\nu_\theta(s_0)\big] + \lambda^\top \hat{c}$$

where $\hat{w}(s,a,s') := (f')^{-1}\big(\frac{1}{\alpha}(R(s,a) - \lambda^\top C(s,a) + \gamma \nu_\theta(s') - \nu_\theta(s))\big)_+$. While $J_\nu^\lambda$ can be a biased estimate of $L(\lambda, \nu)$ in (12) in general due to $(f')^{-1}(\mathbb{E}[\cdot]) \neq \mathbb{E}[(f')^{-1}(\cdot)]$, we can show that $J_\nu^\lambda$ is an upper bound of $L(\nu, \lambda)$ (i.e. we minimize the upper bound), and $J_\nu^\lambda = L(\nu, \lambda)$ holds if transition dynamics are deterministic (e.g. Mujoco control tasks) (Lee et al., 2021). The parameters of the $\chi_\phi$ network and $\tau$ can be trained by:

$$\min_{\tau \geq 0, \phi} \sum_{k=1}^{K} \tau \log \mathbb{E}_{x \sim d^D}\Big[ \exp\big(\tfrac{1}{\tau}(\hat{w}(s,a,s')(C_k(s,a) + \gamma \chi_{\phi,k}(s') - \chi_{\phi,k}(s)) \qquad (24)$$

$$+ (1-\gamma)\chi_{\phi,k}(s_0))\big)\Big] + \tau_k \epsilon$$

This involves a logarithm outside of the expectation, which implies that mini-batch approximations would introduce a bias. Still, we adopt the simple mini-batch approximation for computational efficiency, with a moderately large batch size (e.g. 1024), which worked well in practice. The empirical form of the loss we use is given by:

$$\min_{\tau \geq 0, \phi} J_{\tau,\chi}(\tau, \phi) = \mathbb{E}_{\text{batch}(D) \sim D}\Big[ \sum_{k=1}^{K} \tau \log \mathbb{E}_{x \sim \text{batch}(D)}\big[ \exp\big(\tfrac{1}{\tau}(\hat{w}(s,a,s') \cdot \qquad (25)$$

$$(C_k(s,a) + \gamma \chi_{\phi,k}(s') - \chi_{\phi,k}(s)) + (1-\gamma)\chi_{\phi,k}(s_0)))\big]\Big] + \tau_k \epsilon\Big]$$

Lastly, $\lambda$ and the policy parameter $\psi$ are optimized by:

$$\min_{\lambda \geq 0} J_\lambda(\lambda) = \lambda^\top (\hat{c} - J_{\tau,\chi}(\tau, \phi)) \tag{26}$$

$$\min_\psi J_\pi(\psi) = -\mathbb{E}_{x \sim d^D}[\hat{w}(s, a, s') \log \pi_\psi(a|s)] \tag{27}$$

The complete pseudo-code is described in Appendix C, where every parameter is optimized jointly.

## 4    EXPERIMENTS

### 4.1    TABULAR CMDPs (RANDOMLY GENERATED CMDPs)

We first probe how COptiDICE can improve the reward performance beyond the data-collection policy while satisfying the given cost constraint via repeated experiments. We follow a protocol similar to the random MDP experiment in (Laroche et al., 2019; Lee et al., 2021) but with cost as an additional consideration. We conduct repeated experiments for 10K runs, and for each run, a CMDP $M$ is generated randomly with the cost threshold $\hat{c} = 0.1$. We test with two types of data-collection policy $\pi_D$, a constraint-satisfying policy (i.e. $V_C(\pi_D) = 0.09$) and a constraint-violating policy (i.e. $V_C(\pi_D) = 0.11$). Then, a varying number of trajectories $N \in \{10, 20, 50, 100, 200, 500, 1000, 2000\}$ are collected from the sampled CMDP using the constructed data-collection policy $\pi_D$, which constitutes the offline dataset $D$. Finally, the offline dataset $D$ is given to each offline constrained RL algorithm, and its reward and cost performance is evaluated. For the reward, we evaluate the normalized performance of the policy $\pi$ by $\frac{V_R(\pi) - V_R(\pi_D)}{V_R(\pi^*) - V_R(\pi_D)} \in (-\infty, 1]$ to see the performance improvement of $\pi$ over $\pi_D$ intuitively, where $\pi^*$ is the optimal policy of the underlying CMDP. More details can be found in Appendix E.

Offline *constrained* RL has been mostly unexplored, thus lacks published baseline algorithms. We consider the following three baselines. First, `BC` denotes the simple behavior cloning algorithm to see whether the proposed method is just remembering the dataset. Second, `Baseline` denotes the algorithm that constructs a maximum-likelihood estimation (MLE) CMDP $\hat{M}$ using $D$ and then solves the MLE CMDP using a tabular CMDP solver (LP solver) (Altman, 1999). Third, `C-SPIBB` is the variant of SPIBB (an offline RL method for tabular MDPs) (Laroche et al., 2019), where we modified SPIBB to deal with the cost constraint by Lagrange relaxation (Appendix E).

Figure 1a presents the result when the data-collection policy is constraint-satisfying. The performance of BC approaches the performance of the $\pi_D$ as the size of the dataset increases, as expected. When the size of the offline dataset is very small, Baseline severely violates the cost constraint when its computed policy is deployed to the real environment (cost red curve in Figure 1a), and it even fails to improve the reward performance over $\pi_D$ (reward red curve in Figure 1a). This result is expected since Baseline overfits to the MLE CMDP, exploiting the highly uncertain state-actions. This can cause a significant gap between the stationary distribution of the optimized policy computed in $M$ and the one computed in $\hat{M}$, leading to failure in both reward performance improvement and cost constraint satisfaction. To prevent such distributional shift, offline RL algorithms commonly adopt the pessimism principle, encouraging staying close to the data support. We can observe that such pessimism principle is also effective in offline constrained RL: both C-SPIBB (orange) and the naive version of COptiDICE (green) show consistent policy improvement over the data-collection policy while showing better constraint satisfaction. Still, the pessimism principle alone is not sufficient to ensure constraint satisfaction, raising the need for additional treatment for cost-conservativeness. Finally, our COptiDICE (blue) shows much stricter cost satisfaction than other baseline algorithms while outperforming Baseline and C-SPIBB in terms of reward performance.

Figure 1b presents the result when the data-collection policy is constraint-violating, where all the baseline algorithms exhibit severe cost violation. This result shows that if the agent is encouraged to stay close to the constraint-violating policy, it may negatively affect the constraint satisfaction, although the pessimism principle was beneficial in terms of reward maximization. In this situation, only COPtiDICE (blue) could meet the given constraint in general, which demonstrates the effectiveness of our proposed method of constraining the upper bound of cost value. Although COptiDICE (blue) shows reward performance degradation when the size of dataset is very small, this is natural in that it sacrifices the reward performance to lower the cost value to meet the constraint conservatively, and it still outperforms the baseline in this low-data regime.

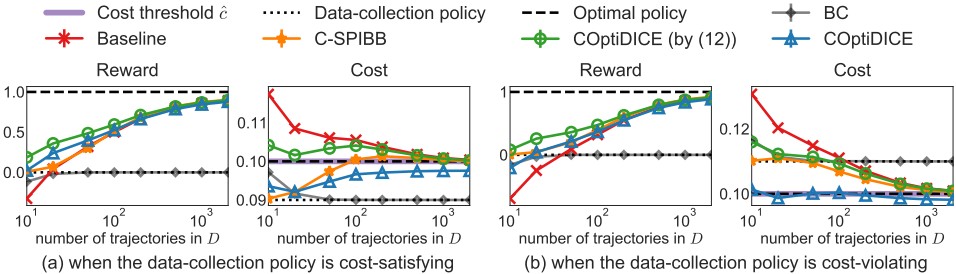

Figure 1: Result of tabular COptiDICE and baseline algorithms in random tabular CMDPs for the varying number of trajectories and two types of data-collection policies. Plots of (a) correspond to the case when the data-collection policy is constraint-satisfying, while the data-collection policy is constraint-violating in the last two plots of (b). The mean of normalized reward performance and the mean of cost value are reported for 10K runs, where the error bar denotes the standard error.

## 4.2 CONTINUOUS CONTROL TASKS (RWRL SUITE)

We also evaluate COptiDICE on domains from the Real-World RL (RWRL) suite (Dulac-Arnold et al., 2020), where the safety constraints are employed. The cost of 1 is given if the task-specific safety constraint is violated at each time step, and the goal is to compute a policy that maximizes rewards while bounding the average cost up to $\hat{c}$. Per-domain safety constraints and the cost constraint thresholds are given in Appendix E. Due to lack of an algorithm that addresses offline constrained RL in continuous action space, we compare COptiDICE with simple baseline algorithms, i.e. `BC`: A simple behavior-cloning agent, `CRR` (Wang et al., 2020): a state-of-the-art (unconstrained) offline RL algorithm, and `C-CRR`: the constrained variant of CRR with Lagrangian relaxation where a cost critic and a Lagrange multiplier for the cost constraint are introduced (Appendix E).

Since there is no standard dataset for offline constrained RL, we collected data using online constrained RL agents (C-DMPO; the constrained variant of Distributional MPO) (Abdolmaleki et al., 2018; Mankowitz et al., 2021). We trained the online C-DMPO with various cost thresholds $\hat{c}$ and saved checkpoints at regular intervals, which constitutes the pool of policy checkpoints. Then, we generated datasets, where each of them consists of $\beta \times 100\%$ constraint-satisfying trajectories and the rest constraint-violating trajectories. The trajectories were sampled by policies in the pool of policy checkpoints. Since there is a trade-off between reward and cost in general, the dataset is a mixture of low-reward-low-cost trajectories and high-reward-high-cost trajectories.

Figure 2 presents our results in RWRL tasks, where the dataset contains mostly constraint-satisfying trajectories ($\beta = 0.8$), with some cost-violating trajectories. This type of dataset is representative of many practical scenarios: the data-collecting agent behaves safely in most cases, but sometimes exhibit exploratory behaviors which can be leveraged for potential performance improvement. Due to the characteristics of these datasets, BC (orange) generally yields constraint-satisfying policy, but its reward performance is also very low. CRR (red) significantly improves reward performance over BC, but it does not ensure that the constraints are satisfied. C-CRR (brown) incurs relatively lower cost value than BC in Walker, Quadruped, and Humanoid, but its reward performance is clearly worse than BC. The naive COptiDICE algorithm (green) takes the cost constraint into account but nevertheless frequently violates the constraint due to limitations of OPE. Finally, COptiDICE (blue) computes a policy that is more robust to cost violation than other algorithms, while significantly outperforming BC in terms of reward performance. We observe that the hyperparameter $\epsilon$ in (16) controls the degree of cost-conservativeness as expected: larger values of $\epsilon$ lead to overestimates of the cost value, yielding a more conservative policy.

To study the dependence on the characteristics of the dataset we experiment with different values of $\beta$. Figure 3a show the result for $\beta = 0.8$ (low-reward-low-cost data), Figure 3b for $\beta = 0.5$, and Figure 3c for $\beta = 0.2$ (high-reward-high-cost data). These results show the expected trend: more high-reward-high-cost data leads to a joint increase in rewards and costs of all agents. A simple modification of the unconstrained CRR to the constrained one (brown) was not effective enough to satisfy the constraint. Our vanilla offline constrained RL algorithm (green), encouraged to stay close to the data, also suffers from severe constraint violation when most of the trajectories are given as the constraint-violating ones, which is similar to the result of Figure 1c-1d. Finally,

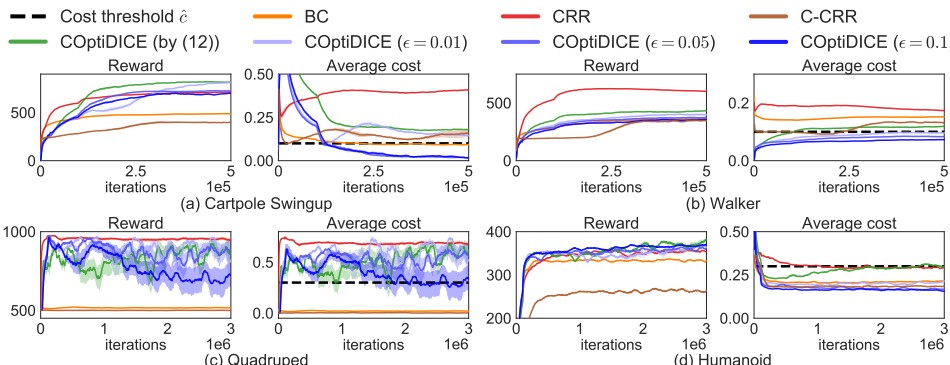

Figure 2: Result of RWRL control tasks. For each task, we report the reward return and the average cost. The results are averaged over 5 runs, and the shaded area represents the standard error.

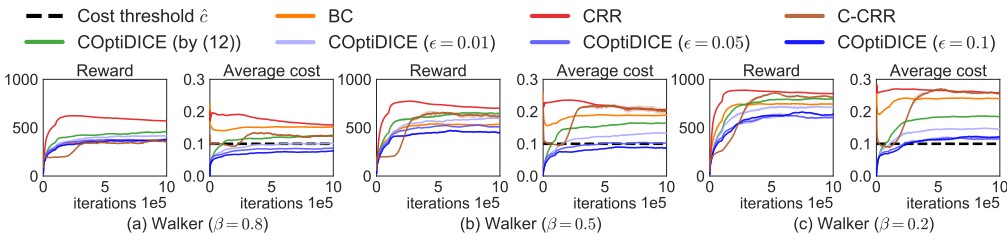

Figure 3: Result on RWRL walker using three dataset configurations with different levels of constraint satisfaction $\beta$: for (a) the data is obtained with $\beta = 0.8$ (low-reward-low-cost data), (b) with $\beta = 0.5$, and (c) with $\beta = 0.2$ (high-reward-high-cost data).

COptiDICE (blue) demonstrates more robust behavior to avoid constraint violations across dataset configurations, highlighting the effectiveness of our method constraining the cost upper bound.

## 5 DISCUSSION AND CONCLUSION

The notion of safety in RL has been captured in various forms such as risk-sensitivity (Chow et al., 2015; Urpí et al., 2021; Yang et al., 2021), Robust MDP (Iyengar, 2005; Tamar et al., 2014), and Constrained MDP (Altman, 1999), among which we focus on CMDP as it provides a natural formalism to encode safety specifications (Ray et al., 2019). Most of the existing constrained RL algorithms (Achiam et al., 2017; Tessler et al., 2019; Satija et al., 2020) are on-policy algorithms, which cannot be applied to the offline setting directly. A recent exception is the work by Le et al. (2019) that also aims to solve constrained RL in an offline setting, though its method is limited to discrete action spaces and relies on solving an MDP completely as an inner optimization, which is inefficient. It also relies on the vanilla OPE estimate of the policy cost, which could result in severe constraint violation when deployed. Lastly, in a work done concurrently to ours, Xu et al. (2021) also exploits overestimated cost value to deal with the cost constraint, but their approach relies on an actor-critic algorithm, while ours relies on stationary distribution optimization.

In this work, we have presented a DICE-based offline constrained RL algorithm, COptiDICE. DICE-family algorithms have been proposed for off-policy evaluation (Nachum et al., 2019a; Zhang et al., 2020a;b; Yang et al., 2020b; Dai et al., 2020), imitation learning (Kostrikov et al., 2019), offline policy selection (Yang et al., 2020a), and RL (Nachum et al., 2019b; Lee et al., 2021), but none of them is for *constrained* RL. Our first contribution was a derivation that constrained offline RL can be tackled by solving a single minimization problem. We demonstrated that such approach, in its simplest form, suffers from constraint violation in practice. To mitigate the issue, COptiDICE instead constrains the cost upper bound, which is estimated in a way that exploits the distribution correction $w$ obtained by solving the RL problem. Such reuse of $w$ eliminates the nested optimization in CoinDICE (Dai et al., 2020), and COptiDICE can be optimized efficiently as a result. Experimental results demonstrated that our algorithm achieved better trade-off between reward maximization and constraint satisfaction than several baselines, across domains and conditions.

ACKNOWLEDGMENTS

The authors would like to thank Rui Zhu for technical support and Sandy Huang for paper feedback. Kee-Eung Kim was supported by the National Research Foundation (NRF) of Korea (NRF-2019R1A2C1087634, NRF-2021M3I1A1097938) and the Ministry of Science and Information communication Technology (MSIT) of Korea (IITP No.2019-0-00075, IITP No.2020-0-00940, IITP No.2021-0-02068).

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

## A  ALGORITHM FOR UNDISCOUNTED CMDP

For $\gamma = 1$, the optimization problem (4-7) should be modified by adding an additional normalization constraint $\sum_{s,a} d(s,a) = 1$.

$$\max_d \mathbb{E}_{(s,a) \sim d}[R(s,a)] - \alpha D_f(d||d^D) \tag{28}$$

$$\text{s.t. } \mathbb{E}_{(s,a) \sim d}[C_k(s,a)] \leq \hat{c}_k \qquad \forall k = 1, \dots, K \tag{29}$$

$$\sum_{a'} d(s',a') = (1-\gamma)p_0(s') + \gamma \sum_{s,a} d(s,a)T(s'|s,a) \qquad \forall s' \tag{30}$$

$$d(s,a) \geq 0 \qquad \forall s,a \tag{31}$$

$$\sum_{s,a} d(s,a) = 1 \tag{32}$$

Then, we consider the Lagrangian:

$$\min_{\lambda \geq 0, \nu, \mu} \max_{d \geq 0} \mathbb{E}_{(s,a) \sim d}[R(s,a)] - \alpha D_f(d||d^D) - \sum_{k=1}^{K} \lambda_k \big( \mathbb{E}_{(s,a) \sim d}[C_k(s,a)] - \hat{c}_k \big)$$

$$- \sum_{s'} \nu(s') \Big[ \sum_{a'} d(s',a') - (1-\gamma)p_0(s') - \gamma \sum_{s,a} d(s,a)T(s'|s,a) \Big] - \mu \Big[ \sum_{s,a} d(s,a) - 1 \Big] \tag{33}$$

where $\mu \in \mathbb{R}$ is the Lagrange multiplier for the normalization constraint (32). Then, we rearrange the terms so that the direct dependence on $d$ is eliminated, while introducing new optimization variables $w(s,a) = \frac{d(s,a)}{d^D(s,a)}$ that represent stationary distribution corrections:

$$\min_{\lambda \geq 0, \nu, \mu} \max_{d \geq 0} \mathbb{E}_{\substack{(s,a) \sim d \\ s' \sim T(s,a)}} \big[ R(s,a) - \lambda^\top C(s,a) + \gamma \nu(s') - \nu(s) - \mu \big] - \alpha \mathbb{E}_{(s,a) \sim d^D} \Big[ f\Big( \tfrac{d(s,a)}{d^D(s,a)} \Big) \Big]$$

$$+ (1-\gamma) \mathbb{E}_{s_0 \sim p_0}[\nu(s_0)] + \lambda^\top \hat{c} + \mu$$

$$= \min_{\lambda \geq 0, \nu, \mu} \max_{w \geq 0} \mathbb{E}_{(s,a) \sim d^D} \big[ w(s,a)(e_{\lambda,\nu}(s,a) - \mu) - \alpha f(w(s,a)) \big]$$

$$+ (1-\gamma) \mathbb{E}_{s_0 \sim p_0}[\nu(s_0)] + \lambda^\top \hat{c} + \mu \tag{34}$$

which yields the minmax optimization problem that can be optimized in a fully offline manner. Due to the strict convexity of $f$, we can also derive the closed form solution of the inner maximization problem:

$$w^*_{\lambda,\nu,\mu}(s,a) = (f')^{-1} \Big( \tfrac{1}{\alpha}(e_{\lambda,\nu}(s,a) - \mu) \Big)_+ \tag{35}$$

which simplify the the minmax problem (34) into the following single minimization problem:

$$\min_{\lambda \geq 0, \nu, \mu} L(\lambda, \nu, \mu) = \mathbb{E}_{(s,a) \sim d^D} \big[ w^*_{\lambda,\nu,\mu}(s,a)(e_{\lambda,\nu}(s,a) - \mu) - \alpha f(w^*_{\lambda,\nu,\mu}(s,a)) \big]$$

$$+ (1-\gamma) \mathbb{E}_{s_0 \sim p_0}[\nu(s_0)] + \lambda^\top \hat{c} + \mu \tag{36}$$

Once the optimal solution $(\lambda^*, \nu^*, \mu^*)$ are obtained, we can also compute the stationary distribution corrections of the optimal policy using (35). Still, naively using (36) would similarly result in cost violation when the resulting policy is deployed to the real environment. We therefore adopt constraining the upper bound of cost value:

$$\nu, \mu \leftarrow \underset{\nu, \mu}{\arg\min}\, L(\lambda, \nu, \mu) \qquad \text{(RL for } R - \lambda^\top C \text{ using (36))} \tag{37}$$

$$\tau, \chi \leftarrow \underset{\tau \geq 0, \chi}{\arg\min} \sum_{k=1}^{K} \ell_k(\tau_k, \chi_k; w^*_{\lambda,\nu,\mu}) \qquad \text{(Upper cost value estimation using (18))}$$

$$\lambda \leftarrow \underset{\lambda \geq 0}{\arg\min}\, \lambda^\top \big( \hat{c} - \underbrace{\ell(\tau, \chi; w^*_{\lambda,\nu,\mu})}_{\approx \text{UpperBound}(\hat{V}_C(\pi))} \big) \qquad \text{(Cost Lagrange multiplier)}$$

which completes the brief description of COptiDICE for $\gamma = 1$.

## B  PROOFS

**Proposition 1.** *For any $\nu$ and $\lambda$, the closed-form solution of*

$$\max_{w \geq 0} L(w, \lambda, \nu) := \mathbb{E}_{(s,a) \sim d^D} \left[ w(s,a) e_{\lambda,\nu}(s,a) - \alpha f(w(s,a)) \right] + (1-\gamma)\mathbb{E}_{s_0 \sim p_0}[\nu(s_0)] + \lambda^\top \hat{c}$$

(38)

*is given by:*

$$w^*_{\lambda,\nu}(s,a) = (f')^{-1} \left( \tfrac{1}{\alpha} e_{\lambda,\nu}(s,a) \right)_+ \text{ where } x_+ = \max(0,x)$$

(39)

*Proof.* For a fixed $\lambda$ and $\nu$, we write the dual of the maximization problem $\max_{w \geq 0} L(w, \lambda, \nu)$:

$$\max_w \min_{\kappa \geq 0} L(w, \lambda, \nu) + \sum_{s,a} \kappa(s,a) w(s,a)$$

$$\Leftrightarrow \max_w \min_{\kappa \geq 0} L(w, \lambda, \nu) + \sum_{s,a} \kappa(s,a) d^D(s,a) w(s,a) \quad (\because d^D > 0).$$

By the strong duality, it is sufficient to consider KKT condition for $(w^*, \kappa^*)$.

*Condition 1 (Primal feasibility)* $w^*(s,a) \geq 0 \; \forall s,a$

*Condition 2 (Dual feasibility)* $\kappa^*(s,a) \geq 0 \; \forall s,a$

*Condition 3 (Stationarity)* $d^D(s,a)(e_{\lambda,\nu}(s,a) - \alpha f'(w^*(s,a)) + \kappa^*(s,a)) = 0 \; \forall s,a$

$$\Leftrightarrow f'(w^*(s,a)) = \tfrac{1}{\alpha}(e_{\lambda,\nu}(s,a) + \kappa^*(s,a))$$

$$\Leftrightarrow w^*(s,a) = (f')^{-1} \left( \tfrac{1}{\alpha}(e_{\lambda,\nu}(s,a) + \kappa^*(s,a)) \right)$$

(40)

*Condition 4 (Complementary slackness)* $w^*(s,a)\kappa^*(s,a) = 0 \; \forall s,a$

Then, we will show that:

$$w^*_{\lambda,\nu}(s,a) = (f')^{-1} \left( \tfrac{1}{\alpha} e_{\lambda,\nu}(s,a) \right)_+$$

(41)

satisfies the KKT conditions for all $(s,a)$. First, *Primal feasibility* is always satisfied by definition of $w^*_{\lambda,\nu}(s,a)$. Then, we consider either $\tfrac{1}{\alpha} e_{\lambda,\nu}(s,a) > f'(0)$ or $\tfrac{1}{\alpha} e_{\lambda,\nu}(s,a) \leq f'(0)$ for each $(s,a)$.

$\left( \text{Case 1: } \tfrac{1}{\alpha} e_{\lambda,\nu}(s,a) > f'(0) \right)$: In this case, $\kappa^*(s,a) = 0$, where *Dual feasibility* and *Complementary slackness* is satisfied. *Stationarity* also holds by:

$$\begin{aligned} w^*_{\lambda,\nu}(s,a) &= (f')^{-1} \left( \tfrac{1}{\alpha} e_{\lambda,\nu}(s,a) \right)_+ \\ &= (f')^{-1} \left( \tfrac{1}{\alpha} e_{\lambda,\nu}(s,a) \right) && \text{(by assumption)} \\ &= (f')^{-1} \left( \tfrac{1}{\alpha}(e_{\lambda,\nu}(s,a) + \kappa^*(s,a)) \right) && \Leftrightarrow (40) \end{aligned}$$

Therefore, KKT conditions (Conditions 1-4) are satisfied.

$\left( \text{Case 2: } \tfrac{1}{\alpha} e_{\lambda,\nu}(s,a) \leq f'(0) \right)$: In this case, $\kappa^*(s,a) = \alpha f'(0) - e_{\lambda,\nu}(s,a)$, where *Dual feasibility* holds by assumption. Also, $w^*_{\lambda,\nu}(s,a) = 0$ by assumption, which implies *Complementary slackness* also holds. Finally, *Stationarity* also holds by:

$$\begin{aligned} w^*_{\lambda,\nu}(s,a) &= (f')^{-1} \left( \tfrac{1}{\alpha} e_{\lambda,\nu}(s,a) \right)_+ \\ &= 0 && \text{(by assumption)} \\ &= (f')^{-1} \left( f'(0) \right) \\ &= (f')^{-1} \left( \tfrac{1}{\alpha}(e_{\lambda,\nu}(s,a) + \kappa^*(s,a)) \right) && \Leftrightarrow (40) \end{aligned}$$

As a consequence, KKT conditions (Conditions 1-4) are always satisfied with $w^*_{\lambda,\nu}(s,a) = (f')^{-1} \left( \tfrac{1}{\alpha} e_{\lambda,\nu}(s,a) \right)_+$, which concludes the proof. $\qquad\square$

**Proposition 2.** *The constrained optimization problem (15-17) can be reduced to solving the following unconstrained minimization problem:*

$$\min_{\tau \geq 0, \chi} \ell_k(\tau, \chi; w) = \tau \log \mathbb{E}_{x \sim d^D} \Big[ \exp \Big( \tfrac{1}{\tau} \big( w(s,a)(C_k(s,a) + \gamma\chi(s') - \chi(s)) + (1-\gamma)\chi(s_0) \big) \Big) \Big] + \tau\epsilon$$

(18)

*where $\tau \in \mathbb{R}_+$ corresponds to the Lagrange multiplier for the constraint (16), and $\chi(s) \in \mathbb{R}$ corresponds to the Lagrange multiplier for the constraint (17). In other words, $\min_{\tau \geq 0, \chi} \ell(\tau, \chi) = \mathbb{E}_{(s,a) \sim \tilde{p}^*}[w(s,a)C_k(s,a)]$ where $\tilde{p}^*$ is the optimal perturbed distribution of the problem (15-17). Also, for the optimal solution $(\tau^*, \chi^*)$, $\tilde{p}^*$ is given by:*

$$\tilde{p}^*(x) \propto d^D(x) \underbrace{\exp \Big( \tfrac{1}{\tau^*} \big( w(s,a)(C_k(s,a) + \gamma\chi^*(s') - \chi^*(s)) + (1-\gamma)\chi^*(s_0) \big) \Big)}_{=: \ \omega^*(x) \ \text{(unnormalized weight for } x = (s_0, s, a, s'))}$$

(19)

*Proof.* For the given constrained optimization problem:

$$\max_{\tilde{p} \in \Delta(X)} \mathbb{E}_{(s_0,s,a,s') \sim \tilde{p}}[w(s,a)C_k(s,a)]$$

(42)

$$\text{s.t. } D_{\mathrm{KL}}(\tilde{p}(s_0, s, a, s') || d^D(s_0, s, a, s')) \leq \epsilon$$

(43)

$$\sum_{a'} \tilde{p}(s', a')w(s', a') = (1-\gamma)\tilde{p}_0(s') + \gamma \sum_{s,a} \tilde{p}(s,a)w(s,a)\tilde{p}(s'|s,a) \quad \forall s'$$

(44)

We consider the Lagrangian:

$$\min_{\tau \geq 0, \chi, \zeta} \max_{\tilde{p} \geq 0} \sum_x \tilde{p}(s_0, s, a, s')[w(s,a)C_k(s,a)] - \tau \Big( \sum_x \tilde{p}(s_0, s, a, s') \Big[ \log \tfrac{\tilde{p}(s_0,s,a,s')}{d^D(s_0,s,a,s')} \Big] - \epsilon \Big)$$

$$- \sum_{s'} \chi(s') \Big[ \sum_{a'} \tilde{p}(s', a')w(s', a') - (1-\gamma)\tilde{p}_0(s') - \gamma \sum_{s,a} \tilde{p}(s,a)w(s,a)\tilde{p}(s'|s,a) \Big]$$

$$- \zeta \Big[ \sum_x \tilde{p}(s_0, s, a, s') - 1 \Big]$$

(45)

where $\tau \in \mathbb{R}_+$ is the Lagrange multiplier for KL constraint (43), $\chi(s') \in \mathbb{R}$ is the Lagrange multiplier for (44), and $\zeta \in \mathbb{R}$ is the Lagrange multiplier for the normalization constraint that ensures $\sum_x \tilde{p}(x) = 1$. Then, we rearrange (45) by:

$$\min_{\tau \geq 0, \chi, \zeta} \max_{\tilde{p} \geq 0} \sum_x \tilde{p}(x) \Big[ w(s,a)\big(C_k(s,a) + \gamma\chi(s') - \chi(s)\big) + (1-\gamma)\chi(s_0) - \tau \log \tfrac{\tilde{p}(x)}{d^D(x)} \Big]$$

$$+ \tau\epsilon - \zeta \Big[ \sum_x \tilde{p}(x) - 1 \Big] =: g(\tau, \chi, \zeta, \tilde{p})$$

(46)

Then, we can compute the non-parametric closed form solution for each sample $x = (s_0, s, a, s')$ for the inner-maximization problem. Thanks to convexity of KL-divergence, it is sufficient to consider $\frac{\partial g(\tau, \chi, \zeta, \tilde{p})}{\partial \tilde{p}(x)} = 0$ for each $x$. Then,

$$\frac{\partial g(\tau, \chi, \zeta, \tilde{p})}{\partial \tilde{p}(x)} = w(s,a)\big(C_k(s,a) + \gamma\chi(s') - \chi(s)\big) + (1-\gamma)\chi(s_0) - \tau \log \tfrac{\tilde{p}(x)}{d^D(x)} + \tau - \zeta = 0$$

$$\Rightarrow \tilde{p}(x) \propto d^D(x) \exp \Big( \tfrac{1}{\tau} \big( w(s,a)\big(C_k(s,a) + \gamma\chi(s') - \chi(s)\big) + (1-\gamma)\chi(s_0) \big) \Big)$$

(47)

with some normalization constant that ensures $\sum_x \tilde{p}(x) = 1$, which is described with respect to $\zeta$. Then, by plugging (47) into (46), we obtain the result. Also, (19) is the direct result of (47).

$\square$

## C  PSEUDOCODE OF COPTiDICE

---

**Algorithm 1** COptiDICE

---

**Input:** An offline dataset $D = \{(s_0, s, a, r, c, s')_i\}_{i=1}^N$, a learning rate $\eta$.
1: Initialize parameter vectors $\theta, \phi, \lambda, \tau, \psi$.
2: **for** each gradient step **do**
3:    Sample mini-batches from $D$.
4:    Compute gradients and perform SGD update:
5:      $\theta \leftarrow \theta - \eta\nabla_\theta J_\nu(\theta)$       (Eq. (23))      $\phi \leftarrow \phi - \eta\nabla_\phi J_{\tau,\chi}(\tau, \phi)$  (Eq. (25))
6:      $\tau \leftarrow \left[\tau - \eta\nabla_\tau J_{\tau,\chi}(\tau, \phi)\right]_+$  (Eq. (25))      $\lambda \leftarrow \left[\lambda - \eta J_\lambda(\lambda)\right]_+$      (Eq. (26))
7:      $\psi \leftarrow \psi - \eta\nabla_\psi J_\pi(\psi)$       (Eq. (27))
8: **end for**

---

## D  COMPARISON WITH COINDICE

CoinDICE (Dai et al., 2020) is a DICE-family algorithm for off-policy confidence interval estimation. For a given policy $\pi$, CoinDICE essentially solves the following constrained optimization problem to estimate the upper confidence interval of the cost value:

$$\max_{\tilde{p} \in \Delta(X)} \max_{w \geq 0} \mathbb{E}_{(s_0,s,a,s') \sim \tilde{p}}[w(s,a)C_k(s,a)] \tag{48}$$

$$\text{s.t. } D_{\text{KL}}(\tilde{p}(s_0, s, a, s') || d^D(s_0, s, a, s')) \leq \epsilon \tag{49}$$

$$\tilde{p}(s', a')w(s', a') = (1 - \gamma)\tilde{p}_0(s')\pi(a'|s') + \gamma\sum_{s,a}\tilde{p}(s,a)w(s,a)\tilde{p}(s'|s,a)\pi(a'|s') \quad \forall s', a' \tag{50}$$

The constraint (50) is analogous to the $\pi$-dependent Bellman flow constraint:

$$d^\pi(s', a') = (1 - \gamma)p_0(s')\pi(a'|s') + \gamma\sum_{s,a}d^\pi(s,a)T(s'|s,a)\pi(a'|s') \quad \forall s', a' \tag{51}$$

It is well known that the transposed Bellman equation (51) always has a unique solution $d^\pi$, i.e. the stationary distribution of the given policy $\pi$. Note that for a fixed $\tilde{p}$, the constrained optimization problem (48-50) is *over-constrained* for $w(s,a)$ by (50), and therefore the optimal solution will simply be given by $w^*(s,a) = \frac{d^\pi(s,a)}{\tilde{p}(s,a)}$ to satisfy the transposed Bellman equation (51) on the empirical MDP defined by $\tilde{p}$. Then, we want to adversarially optimize the distribution $\tilde{p}$ so that it overestimates the cost value by (48). At the same time, we enforce that the distribution $\tilde{p}$ should not be perturbed too much from the empirical dsta distribution $d^D$ by the KL constraint (49). Finally, following the similar derivation in Proposition 2, we can reduce the constrained optimization problem (48-50) to solving the following unconstrained max-min optimization.

$$\max_{w \geq 0} \min_{\tau \geq 0, \nu} \tau \log \mathbb{E}_{\substack{x \sim d^D \\ a_0 \sim \pi(s_0) \\ a' \sim \pi(s')}}\left[\exp\left(\tfrac{1}{\tau}\big(w(s,a)(C_k(s,a) + \gamma\nu(s', a') - \nu(s', a')) + (1 - \gamma)\nu(s_0, a_0))\big)\right)\right] + \tau\epsilon$$

where $\max_{w \geq 0} \min_\nu(\cdot)$ is to estimate $w(s,a) = \frac{d^\pi(s,a)}{\tilde{p}(s,a)}$.

In contrast, we consider the case when $w$ is *given* and aim to solve the following constrained optimization problem.

$$\max_{\tilde{p} \in \Delta(X)} \mathbb{E}_{(s_0,s,a,s') \sim \tilde{p}}[w(s,a)C_k(s,a)] \tag{15}$$

$$\text{s.t. } D_{\text{KL}}(\tilde{p}(s_0, s, a, s') || d^D(s_0, s, a, s')) \leq \epsilon \tag{16}$$

$$\sum_{a'}\tilde{p}(s', a')w(s', a') = (1 - \gamma)\tilde{p}_0(s') + \gamma\sum_{s,a}\tilde{p}(s,a)w(s,a)\tilde{p}(s'|s,a) \quad \forall s' \tag{17}$$

This can be reduced to unconstrained minimization problem, without requiring nested optimization to estimate $w$:

$$\min_{\tau \geq 0, \chi} \tau \log \mathbb{E}_{x \sim d^D}\left[\exp\left(\tfrac{1}{\tau}\big(w(s,a)(C_k(s,a) + \gamma\chi(s') - \chi(s)) + (1 - \gamma)\chi(s_0))\big)\right)\right] + \tau\epsilon$$

# E  EXPERIMENTAL SETTINGS

## E.1  RANDOM CMDPS

For random CMDP experiments, We follow a similar experimental protocol as (Laroche et al., 2019; Lee et al., 2021) with additional consideration of cost constraint.

**Random CMDP generation**  For each run, we constructed a random CMDP with $|S| = 50, |A| = 4, \gamma = 0.95$, and a fixed initial state $s_0$. The transition probability is constructed randomly with connectivity of 4, i.e. for each $(s, a)$, we sample 4 states uniformly, and then, the transition probabilities to those states are determined by $\mathrm{Dirichlet}(1, 1, 1, 1)$. The reward of 1 is given to a single state that minimizes the optimal policy's reward value at $s_0$, and 0 is given anywhere else. This reward design can be roughly understood as choosing a goal state that is most difficult to reach from $s_0$. The cost function is generated randomly, i.e. $C(s, a) \sim \mathrm{Beta}(0.2, 0.2)$ for $\forall s \in S, a \in \{a_2, a_3, a_4\}$ and $C(s, a_1) = 0$ to ensure existence of a feasible policy of the CMDP. Lastly, the cost threshold $\hat{c} = 0.1$ is used.

**Data-collection policy construction**  The pseudo-code for the data-collection policy construction is presented in Algorithm 2, where $M$ is the underlying true CMDP, and $\hat{c}_D \in \{0.09, 0.11\}$ is the hyperparameter that determines the cost value of $\pi_D$. Starting from $\pi_D = \pi^*$, the policy is softened

---

**Algorithm 2** Data-collection policy construction

---

**Input:** CMDP $M = \langle S, A, T, R, C, \hat{c}, p_0, \gamma \rangle$, target cost value of the data-collection policy $\hat{c}_D$

 Compute the optimal policy $\pi^*$ and its reward value function $Q_R^{\pi^*}(s, a)$ on the given CMDP $M$.
 Initialize $\pi_{\mathrm{soft}} \leftarrow \pi^*$ and $\pi_D \leftarrow \pi^*$
 Initialize a temperature parameter $\tau \leftarrow 10^{-6}$
 # Compute 0.9-optimal behavior policy in terms of reward performance.
 **while** $V_R^{\pi_D}(s_0) > 0.9 V_R^{\pi^*}(s_0) + 0.1 V_R^{\pi_{\mathrm{unif}}}(s_0)$ **do**
  Set $\pi_{\mathrm{soft}}$ to $\pi_{\mathrm{soft}}(a|s) \propto \exp\left(\frac{1}{\tau} Q_R^{\pi^*}(s, a)\right) \quad \forall s, a$
  $\pi_D \leftarrow \arg\min_\pi D_f(d^\pi || d^{\pi_{\mathrm{soft}}})$ s.t. $\mathbb{E}_{(s,a) \sim d^\pi}[C(s, a)] \leq \hat{c}_D$
  $\tau \leftarrow \tau / 0.9$
 **end while**
**Output:** The data-collection policy $\pi_D$

---

via $\pi_{\mathrm{soft}}(a|s) \propto \exp(Q_R^*(s, a)/\tau)$ while projecting it into the cost-satisfying one by solving $\pi_D \leftarrow \arg\min_\pi D_f(d^\pi || d^{\pi_{\mathrm{soft}}})$ s.t. $\mathbb{E}_{(s,a) \sim d^\pi}[C(s, a)] \leq \hat{c}_D$. This process is repeated until the reward performance of $\pi_D$ reaches 0.9-optimality while increasing the temperature $\tau$, i.e. $V_R^{\pi_D}(s_0) = 0.9 V_R^{\pi^*}(s_0) + 0.1 V_R^{\pi_{\mathrm{unif}}}(s_0)$.

After the data-collection policy $\pi_D$ is constructed, we sample trajectories using $\pi_D$, which constitutes the offline dataset $D$. We conducted experiments for a varying number of trajectories, i.e. (the number of trajectories) $\in \{10, 20, 50, 100, 200, 500, 1000, 2000\}$. Each episode is terminated either when 50 time steps have reached or the agent reached the goal state that yields a non-zero reward.

**Hyperparameters**  For tabular COptiDICE, we used $\alpha = \frac{1}{N}$ and $\epsilon = \frac{0.1}{N}$, where $N$ denotes the number of trajectories in $D$. We also used $f(x) = \frac{1}{2}(x - 1)^2$, which corresponds to $\chi^2$-divergence.

**C-SPIBB: Constrained variant of SPIBB**  C-SPIBB solves the offline RL problem with respect to $R - \lambda C$ via SPIBB while updating $\lambda$ in the direction of $(\hat{V}_C(\pi_{\mathrm{SPIBB}}) - \hat{c})$, where $\hat{V}_C(\pi_{\mathrm{SPIBB}})$ is evaluated using the MLE CMDP. For C-SPIBB, we used $N_\wedge = 5$ as the hyperparameter of SPIBB.

## E.2 RWRL CONTROL TASKS

**Network architecture and hyperparameters**  We used the ACME framework (Hoffman et al., 2020). For the $\nu_\theta$ network and the $\chi_\phi$ network, we used `LayerNormMLP` with hidden sizes of $[512, 512, 256]$. For the policy network $\pi_\psi$, we used the network architecture used in CRR (Wang et al., 2020). Specifically, we the $\pi_\psi$ network consists of 4 `ResidualMLP` blocks with hidden size of 1024 and a mixture of Gaussians policy head with 5 mixture components. We used the batch size of 1024. We use Adam optimizer with learning rate 3e-4. Similar to ValueDICE (Kostrikov et al., 2021), we additionally adopt gradient penalties (Gulrajani et al., 2017) for the $\nu_\theta$ network and the $\chi_\phi$ network with coefficient 10e-5. We performed grid search for $\alpha \in \{0.01, 0.05, 0.1\}$ for each domain. We used the following $f$ as in OptiDICE (Lee et al., 2021):

$$f(x) = \begin{cases} x \log x - x + 1 & \text{if } 0 < x < 1 \\ \frac{1}{2}(x-1)^2 & \text{if } x \geq 1 \end{cases}$$

**Task specification**  We conduct experiments on domains using the RWRL suite with safety-spec. The *safety coefficient* is a flag in the RWRL suite with safety-spec, and its value can be between 0.0 and 1.0. Lowering the value of the flag incurs more safety-constraint violation (cost of 1). Originally, each domain has multiple types of safety constraints, but we use only one of them, which was the hardest safety constraint to be satisfied by an online constrained RL agent. In summary, we used the following task specifications (safety coefficients, name of the used safety constraint, cost threshold $\hat{c}$).

- Cartpole (realworld-swingup): safety-coeff=0.3, `slider_pos`, $\hat{c} = 0.1$,
- Walker (realworld-walk): safety-coeff=0.3, `joint_velocity_constraint` $\hat{c} = 0.1$.
- Quadruped (realworld-walk): safety-coeff=0.5, `joint_angle_constraint` $\hat{c} = 0.3$.
- Humanoid (realworld-walk): safety-coeff=0.5, `joint_angle_constraint`, $\hat{c} = 0.3$.

The offline dataset consists trajectories of 1000 episodes for Cartpole and Walker, and 5000 episodes for Quadruped and Humanoid.

**C-CRR: Constrained variant of CRR**  C-CRR additionally introduces the cost critic $Q_C$ and the Lagrange multiplier $\lambda$ for the cost constraint. Then, the reward critic and the cost critic are trained by minimizing TD-error for reward and cost respectively. The actor is trained by weighted behavior-cloning: $\max_\pi \mathbb{E}_{(s,a) \sim d^D}[\exp(\hat{A}(s,a)/\beta) \log \pi(a|s)]$ where $\hat{A}(s,a) = (Q_R(s,a) - \lambda Q_C(s,a)) - \frac{1}{m}\sum_{j=1}^{m}(Q_R(s,a^j) - \lambda Q_C(s,a^j))$, with $a^j \sim \pi(\cdot|s)$. This corresponds to optimizing the policy with respect to the scalarized reward $R - \lambda C$. The Lagrange multiplier $\lambda$ is updated in the direction of $(\mathbb{E}_{(s,a) \sim d^D}[Q_C(s,a)] - \hat{c})$.

## F    DISCUSSION ON THE DATASET COVERAGE ASSUMPTION

Although we have made an assumption $d^D > 0$ in Section 3, this is not strictly required for the offline RL algorithm to work in practice. We adopted this assumption same as OptiDICE (Lee et al., 2021) for the simplicity of describing the algorithm derivation from Eq (4-7) to Eq (10): the assumption makes Eq. (10) correspond to solving the *underlying true* CMDP of Eq. (4-7). If we take this assumption off, Eq. (10) then becomes equivalent to solving the *reduced* CMDP where the state and action spaces are limited to the support of $d^D$.

To see this, we consider the following constrained optimization problem, where the optimization variables $d(s, a)$ are defined only for the $(s, a)$ who are within the support of $d^D$. We denote $\hat{p}_0$ and $\hat{T}$ as the empirical initial state distribution and the empirical transition function respectively.

$$\max_{d} \sum_{(s,a)\in\text{Supp}(d^D)} d(s,a)R(s,a) - \alpha D_f(d||d^D) \tag{52}$$

$$\text{s.t.} \sum_{(s,a)\in\text{Supp}(d^D)} d(s,a)[C_k(s,a)] \leq \hat{c}_k \qquad \forall k = 1,\ldots,K$$

$$\sum_{a'\in\text{Supp}(d^D)} d(s',a') = (1-\gamma)\hat{p}_0(s') + \gamma \sum_{(s,a)\in\text{Supp}(d^D)} d(s,a)\hat{T}(s'|s,a) \qquad \forall s' \in \text{Supp}(d^D)$$

$$d(s,a) \geq 0 \qquad \forall s,a \in \text{Supp}(d^D)$$

Then, by considering the Lagrangian for (52) and following the similar derivation of Eq. (8-9), we can finally arrive at Eq. (10) without any assumption on the coverage of $d^D$, due to the fact that $\sum_{(s,a)\in\text{Supp}(d^D)}[d(s,a)(\cdot)] = \sum_{(s,a)\in\text{Supp}(d^D)} d^D(s,a)\frac{d(s,a)}{d^D(s,a)}(\cdot) = \mathbb{E}_{(s,a)\sim d^D}\left[\frac{d(s,a)}{d^D(s,a)}(\cdot)\right]$ always holds. In other words, our offline RL algorithms that rely on Eq. (10) essentially solve the *reduced* CMDP that is limited to the support of $d^D$.

# G  ABLATION EXPERIMENTS ON DIFFERENT COST THRESHOLDS

In order to see the sensitivity of the proposed method to different cost thresholds, we conduct ablation experiments on different cost thresholds using randomly generated CMDPs. The experimental setup is identical to the one described in Section 4.1.

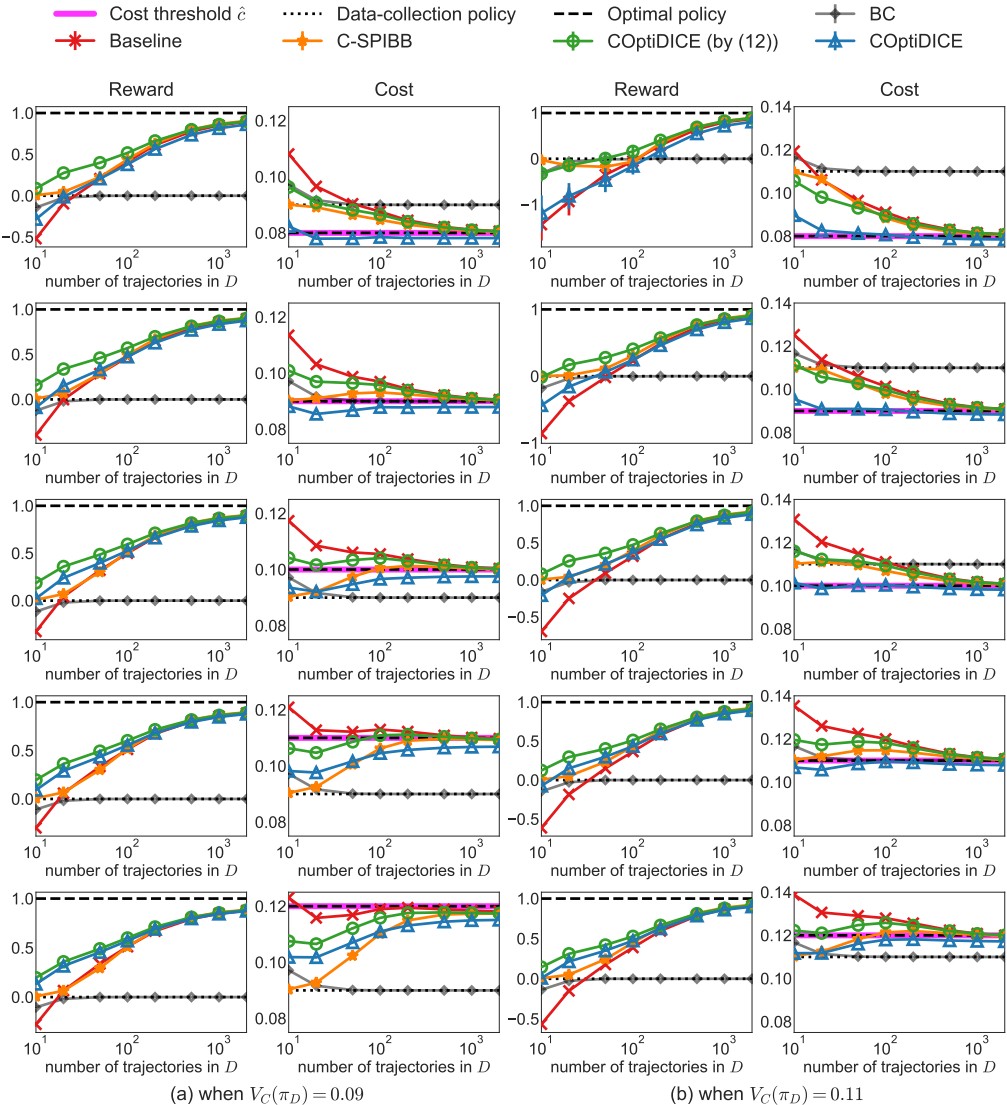

Figure 4: Result of tabular COptiDICE and baseline algorithms in random tabular CMDPs. Plots for the first two columns in (a) correspond to the case when the cost value of the data-collection policy is 0.09 (corresponding to Figure 1a). Plots for the last two columns in (b) correspond to the case when the cost value of the data-collection policy is 0.11 (corresponding to Figure 1b). The mean of normalized reward performance and the mean of cost value are reported for 10K runs, where the error bar denotes the standard error.

On the same data-collection policy (whose cost value is either 0.09 or 0.11) and the same offline dataset as in Figure 1, we tested each algorithm against different target cost thresholds $\hat{c} \in \{0.08, 0.09, 0.10, 0.11, 0.12\}$. Each row in Figure 4 presents the result for each target cost threshold, ranging from $\hat{c} = 0.08$ to $\hat{c} = 0.12$. Our COptiDICE (blue) shows consistent robustness to constraint violation on varying cost thresholds, while other algorithms fail to meet the constraints especially when the threshold is set to low values (e.g. rows 1 and 2).

## H    ADDITIONAL EXPERIMENTS USING MIXTURE DATASET

In Section 4.1, we demonstrated the results when the offline dataset was collected by a single data-collection policy. However, in real-world situations, it would be common that data-collecting agents act safely in most cases but have some unsafe attempts. To simulate this scenario, we conduct additional experiments on the use of a mixture dataset, where the dataset is collected by both constraint-satisfying and constraint-violating policy.

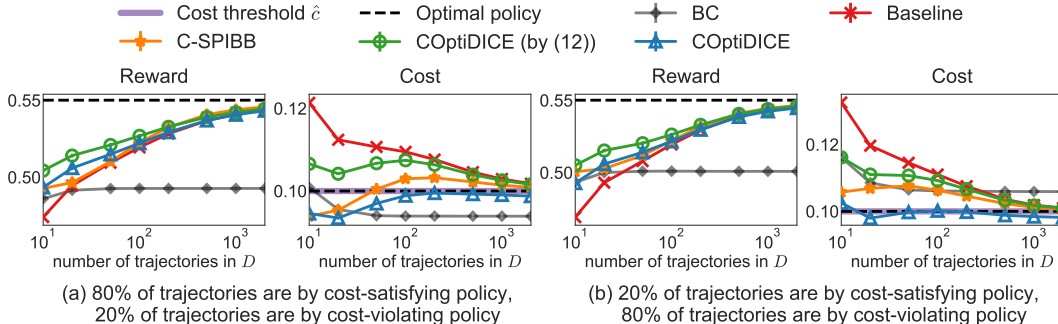

Figure 5: Result of tabular COptiDICE and baseline algorithms in random tabular CMDPs, using mixture dataset. Plots (a) correspond to the case when the dataset is collected generally by the cost-satisfying policy, i.e. 80% of trajectories are by the cost-satisfying policy and 20% are by the cost-violating policy. Plots (b) correspond to the case when the dataset is collected generally by the cost-violating policy, i.e. 20% of trajectories are by the cost-satisfying policy and 80% are by the cost-violating policy. The mean (unnormalized) reward value and the mean cost value are reported for 10K runs, where the error bar denotes the standard error.

Figure 5 presents the result, where the dataset is collected by two policies, the cost-satisfying one (i.e. its cost value is 0.09) and the cost-violating one (i.e. its cost value is 0.11). The overall trend remains the same as in Figure 1 (Figure 5a ≈ Figure 1a, Figure 5b ≈ Figure 1b): when the dataset consists of mostly cost-satisfying trajectories, baseline algorithms exhibits less constraint violation, while they show more constraint violation when the dataset consists of mostly cost-violating ones. Our COptiDICE (blue) still shows much stricter cost satisfaction than other baseline algorithms, demonstrating a better trade-off between reward maximization and constraint satisfaction than baselines.

Furthermore, solving the reduced CMDP is a valid method in the offline RL setting since we may want to optimize the policy only within the dataset support in order to prevent unexpected performance degradation by out-of-distribution actions.

# I  ADDITIONAL EXPERIMENTS ON DATA-COLLECTION POLICY WITH LIMITED EXPLORATION POWER

For the random CMDP experiment, we conducted additional experiments to see the results when the data-collection policy $\pi_D$ does not cover the entire state-action space well. Specifically, we limit the support of $\pi_D(\cdot|s)$: $\pi_D(a|s)$ will have non-zero probabilities only for $a \in \tilde{A}$ where $\tilde{A} \subset A$ is the subset of $A = \{a_1, a_2, a_3, a_4\}$. By doing so, we ensure that $\pi_D$ covers only a part of the entire state-action space. The detailed construction procedure of $\pi_D$ is described in Algorithm 3. We conduct experiments for $\tilde{A} = \{a_1\}$, $\tilde{A} = \{a_1, a_2\}$, and $\tilde{A} = \{a_1, a_2, a_3\}$, and the results are presented in Figure 6.

---

**Algorithm 3** Constructing a data-collection policy with limited action support

---

**Input:** CMDP $M = \langle S, A, T, R, C, \hat{c}, p_0, \gamma \rangle$, the subset of the entire actions, $\tilde{A} \subset A$, to be considered by the data-collection policy, $\hat{c}_D$: the target threshold of the data-collection policy.
 1: **for** each $s \in S$ and $a \in A$ **do**
 2:

$$\tilde{C}(s, a) \leftarrow \begin{cases} C(s, a) & \text{if } a \in \tilde{A} \\ \infty & \text{if } a \notin \tilde{A} \end{cases} \quad \text{\# Eliminate action } a \notin \tilde{A} \text{ by assigning a large cost.}$$

$$\tilde{\pi}_{\text{unif}}(a|s) \leftarrow \begin{cases} 1/|\tilde{A}| & \text{if } a \in \tilde{A} \\ 0 & \text{if } a \notin \tilde{A} \end{cases}$$

 3: **end for**
 4: $\rho \leftarrow 1$
 5: **while** true **do**
 6:   # Artificially lower the cost threshold ($\hat{c}_D \cdot \rho$) until $\pi_D$ meets the target constraint.
 7:   $\tilde{\pi}^* \leftarrow \text{SolveCMDP}(S, A, T, R, \tilde{C}, \hat{c}_D \cdot \rho, p_0, \gamma)$
 8:   $\pi_D \leftarrow 0.9 \cdot \tilde{\pi}^* + 0.1 \cdot \tilde{\pi}_{\text{unif}}$   # To make a stochastic policy within $\tilde{A}$.
 9:   **if** $V_C(\pi_D) > \hat{c}_D$ **then**
10:     $\rho \leftarrow \rho * 0.99$
11:   **else**
12:     break
13:   **end if**
14: **end while**
**Output:** $\pi_D$: the data-collection policy with limited action support.

---

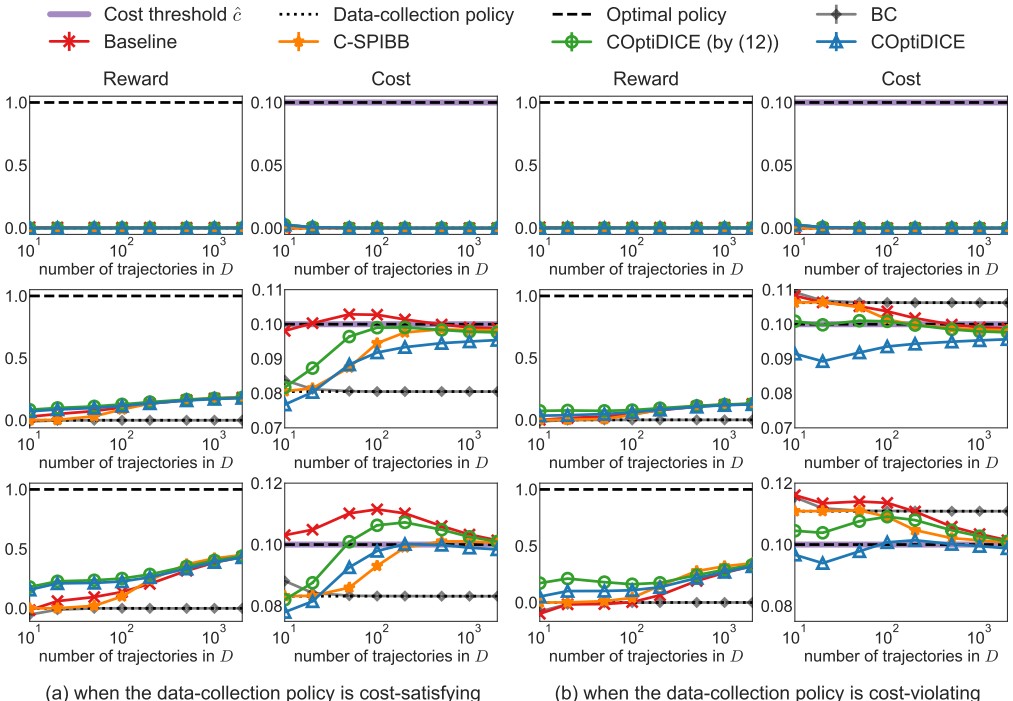

Figure 6: Result of tabular COptiDICE and baseline algorithms in random tabular CMDPs when the data-collection policy has limited exploration power. Plots for the first two columns in (a) correspond to the case when the data-collection policy is cost-satisfying. Plots for the last two columns in (b) correspond to the case when the data-collection policy is cost-violating (except for the first row). The first row denotes the case when $\tilde{A} = \{a_1\}$, the second row denotes the case when $\tilde{A} = \{a_1, a_2\}$, and the third row denotes the case when $\tilde{A} = \{a_1, a_2, a_3\}$. The mean of normalized reward performance and the mean of cost value are reported for 10K runs, where the error bar denotes the standard error.

The first row of Figure 6 shows the result when $\tilde{A} = \{a_1\}$. Since $a_1$ is the zero-cost action as described in Appendix E, the cost value of $\pi_D$ is always 0. Besides, since the dataset contains only a single action $a_1$ for each state due to the *deterministic* data-collection policy, no offline RL algorithms can improve the performance beyond the data-collection policy: there must be more than one action in some states in the dataset, in order for offline RL algorithms to have room to improve the performance in general.

The second row and the third row of Figure 6 shows the result when $\tilde{A} = \{a_1, a_2\}$ and $\tilde{A} = \{a_1, a_2, a_3\}$ respectively. As the size of $\tilde{A}$ increases from 2 to 3, offline RL algorithms further improve the performance over the data-collection policy. This is natural since the larger $\tilde{A}$ implies that there is more room for offline RL algorithms to optimize. Finally, even when the exploration power of the data-collection policy is limited, our COptiDICE (blue) shows a consistent advantage over baseline algorithms, in terms of the better trade-off between reward maximization and constraint satisfaction.

