# OpenReview forum: "COptiDICE: Offline Constrained Reinforcement Learning via Stationary Distribution Correction Estimation"
_ICLR.cc/2022/Conference — ICLR 2022 Spotlight_

### Official Review · Reviewer_qywn · 2021-11-02

**Correctness:** 4
**Technical Novelty And Significance:** 2
**Empirical Novelty And Significance:** 3
**Recommendation:** 6
**Confidence:** 3

**Main Review:**

Strength:
- The paper is well-written and easy to follow. The motivation is well explained.
- The paper considers applying DICE methods to constrained MDP problems. First, by imposing the closed-form solution of $w^*$, the paper reformulates the minimax optimization problem to a single minimization problem. Then,  the paper proposes a practical algorithm for continuous state-action space with neural networks as the function approximation. The paper contributes to the study of CMDP.
- The paper provides an empirical study of the algorithm and shows it outperforms the baseline.

Weakness:
- The assumption that $d^D > 0$ is a strong assumption in offline RL, which assumes that the dataset uniformly covers the whole state-action space. Moreover, since $d^D$ is the empirical distribution of the dataset $D$, such assumption does not hold for continuous state-action space. The paper should provide more discussion on the restrictiveness of such an assumption.
- The algorithm seems to be a combination of OptiDICE and CMDP. By considering the Lagrangian form of CMDP, the problem essentially reduces to an MDP problem. Thus, I expect less novelty of this paper compared with OptiDICE.
- In the experiments, the dataset is generated in a way that enforces its exploration power, which makes the dataset covers the state-action space well. However, in real-world applications, the dataset may not be able to cover the whole state-action space, which is the main challenge of offline RL. In particular, in the case that the dataset does not cover the whole state-action space, I encourage the authors to show if the learned policy can outperform the data collecting policy.




**Summary Of The Paper:**

The paper considers the offline constrained reinforcement learning problem and formulates the problem as a CMDP. First, the paper presents the algorithm COptiDICE, which directly estimates the stationary distribution corrections of the optimal policy. Then, the paper shows that COptiDICE outperforms the baseline algorithms in terms of constraint satisfaction and return-maximization.

**Summary Of The Review:**

Overall, the paper is well-written and motivated. However, due to the lack of discussion on the coverage of the dataset, I think the paper is below the borderline of acceptance. I would be happy to raise my evaluation if the paper could address my concerns in the main review part.

---

> ### Author Response · Authors · 2021-11-22
> **Response to Reviewer qywn**
>
> Thank you very much for your thoughtful feedback.
>
> 1. The assumption $d^D > 0$ is not strictly required for our algorithm to work in practice. We adopted this assumption same as OptiDICE for the simplicity of describing the algorithm derivation from Eq (4-7) to Eq (10): the assumption makes Eq (10) correspond to solving the **underlying true** CMDP of Eq (4-7). If we take this assumption off, Eq (10) then becomes equivalent to solving the **reduced** CMDP where the state and action spaces are limited to the support of $d^D$. Still, solving the reduced CMDP is a valid method in the offline RL setting since we may want to optimize the policy only within the dataset support to prevent severe performance degradation by selecting out-of-distribution actions. More discussions regarding this can be found in Appendix G.
>
>
> 2. We emphasize that our final algorithm is NOT a straightforward combination of OptiDICE and CMDP. We empirically demonstrated that such a simple combination failed to satisfy the cost constraints when deployed to the real environment (green lines in Figure 1-3). To mitigate the challenge of the constraint violation, our COptiDICE constrain the **upper bound** of the cost value, where the cost upper bound is estimated efficiently by solving a **single minimization problem** via reusing the stationary distribution corrections obtained by DICE-based RL. This is in contrast to CoinDICE, where it requires solving a *nested max-min problem* for estimating the cost upper bound (see Eq 18 vs. Eq 20). In conclusion, we would like to highlight that our COptiDICE is not a trivial extension of both OptiDICE and CoinDICE.
>
>
> 3. In Figure 1, the experimental result on random CMDPs demonstrates the case when the dataset does not cover the whole state-action space. For example, in Figure 1a, when the number of trajectories is 50, the dataset covers only about 50% of the entire state-action space, on average. Still, our COptiDICE (blue) significantly outperforms the data-collecting policy in terms of rewards while satisfying the cost constraint.

---

> > ### Comment · Reviewer_qywn · 2021-11-29
> > **Reply**
> >
> > Thank you for your addressing my questions.
> > > In Figure 1, the experimental result on random CMDPs demonstrates the case when the dataset does not cover the whole state-action space. For example, in Figure 1a, when the number of trajectories is 50, the dataset covers only about 50% of the entire state-action space, on average. Still, our COptiDICE (blue) significantly outperforms the data-collecting policy in terms of rewards while satisfying the cost constraint.
> >
> > However, this does not fully answer my question. Although, in this case, the dataset does not cover the entire state-action space, it still contains trajectories that are almost uniformly sampled from the trajectory space. I would expect to see results when the data collecting policy does not have such exploration power.

---

> > > ### Author Response · Authors · 2021-11-30
> > > **Additional experiments**
> > >
> > > Thank you for your reply. We conducted additional experiments on the data-collection policy whose exploration power is explicitly limited. Specifically, we constructed a data-collection policy where its action support contains only a subset of the entire action set.
> > > The detailed construction procedure and its result can be found in the following link (which is anonymized), and this will be included in the final version of the paper.
> > >
> > > https://drive.google.com/file/d/1UDOqiRVFORrZlLaRuOlQ8_5f2gVmaHZG/view?usp=sharing
> > >
> > > Even when the exploration power of the data-collection policy is limited, our COptiDICE shows a consistent advantage over baseline algorithms (including the data-collection policy), in terms of the better trade-off between reward maximization and constraint satisfaction.

---

> > > > ### Comment · Reviewer_qywn · 2021-12-05
> > > > **Reply**
> > > >
> > > > Thank you for your additional experiments. I would raise my score accordingly.

---

### Official Review · Reviewer_jgQ7 · 2021-11-02

**Correctness:** 4
**Technical Novelty And Significance:** 3
**Empirical Novelty And Significance:** 3
**Recommendation:** 8
**Confidence:** 4

**Main Review:**

This paper provide a interesting and novel algorithm to address offline safe RL problems. Most of the theoretical development and empirical looks good to me, but I have the following question for the author.

(1) In order to make sure COptiDICE works in practice, do we need any special requirements for the behavioral policy (sampling distribution)? Specifically, do we required the behavioral policy (or the sampling distribution) to be feasible?

(2) In the experiment of continuous setting, we can still implement CRR as a safe RL method via combining it with primal-dual approach, i.e., you can simply replace the policy optimization step in primal-dual approach with CRR. If we implment such an algorithm, can COptiDICE still outperfrom this "safe-CRR" algorithm?

(3) Has the author considered testing the algorithm in some standard safe RL environment such as Gather, Circle, Half-Cheetah Safe, etc?

**Summary Of The Paper:**

This paper studied the policy optimization problem in the offline constrained MDP setting. Compare with previous works, in which policy gradient based approaches are widely adopted, this paper solves the problem via policy visitation distribution that rooted from the primal-dual formulation of Bellman operator, which is novel. In order to guarantee the constraints are always satisfied, this paper provides a novel approach based on CoinDICE to efficiently estimate an upper bound of constraint violation. The author also provide sufficient empirical verifications to support their proposed algorithm.

**Summary Of The Review:**

The offline safe RL is a very challenging problem, but this paper provide a very promising approach to address several issues in this setting. Specifically, the approach proposed in this paper not only addresses the policy optimization issue in the behavioral agnostic setting but also guarantees the constraints satisfication, which are both significant contributions.

---

> ### Author Response · Authors · 2021-11-22
> **Response to Reviewer jgQ7**
>
> Thank you very much for your thoughtful comments.
>
> 1. For COptiDICE, the behavioral policy (data-collection policy) does not necessarily have to be feasible in terms of constraint satisfaction, as long as the dataset can yield a cost-satisfying policy (e.g. an MLE CMDP constructed by dataset has at least one feasible policy as its solution).
> This is supported by the experimental results in the paper. In Figure 1b, the data-collection policy is indeed violating the constraint (i.e. $V_C(\pi_D) > \hat{c}$), but our COptiDICE (blue) generally computes a cost-satisfying policy while improving reward performance over the data-collection policy as well.
>
>
> 2. We added the baseline algorithms suggested by the reviewer in the experiment. For continuous control tasks (Figure 2-3), we implemented C-CRR (CRR with Lagrangian relaxation to deal with constraints), where a cost critic and a cost Lagrange multiplier are additionally introduced to CRR. C-CRR jointly optimizes actor, reward critic, cost critic, and Lagrange multiplier with four different objective functions. As can be seen from Figure 2-3, our COptiDICE still outperforms C-CRR, showing a better trade-off between reward-maximization and constraint satisfaction.
>
>
> 3. We have chosen the RWRL suite as the set of benchmark environments for our experiments since it has been used both in the context of offline RL [1] and constrained RL [2,3]. Applying our method to other domains such as Gather, Circle, etc. remains as future work.
>
>
> [1] Gulcehre et al., "RL Unplugged: A Suite of Benchmarks for Offline Reinforcement Learning", NeurIPS 2020
>
> [2] Calian et al., "Balancing Constraints and Rewards with Meta-Gradient D4PG", ICLR 2021
>
> [3] Huang et al., "A Constrained Multi-Objective Reinforcement Learning Framework", CoRL 2021

---

> > ### Comment · Reviewer_jgQ7 · 2021-12-01
> > **Thanks for clarification**
> >
> > Thanks for the response. I am glad to raise my score.

---

### Official Review · Reviewer_NsmH · 2021-11-13

**Correctness:** 4
**Technical Novelty And Significance:** 3
**Empirical Novelty And Significance:** 3
**Recommendation:** 8
**Confidence:** 2

**Main Review:**

Strengths

--The paper's approach seems to be very mathematically well-founded. Although the end result is slightly biased, the minibatch-free version simply minimizes a convex optimization problem, so I would assume this bypasses many of the convergence problems common to popular TD-based RL methods.  A line or two clarifying the theoretical guarantees would be nice though.

--The experimental results also seem to demonstrate that the proposed method is better at satisfying the problem constraints than other methods. On a number of environments, this constraint satisficing also comes with little to no cost to overall performance. The authors compare against a good number of baselines on a wide range of environments.

Weaknesses
--While the result is interesting, its novelty seems a bit limited. The transformation from constrained optimization problem to unconstrained optization is of course drawn from previous DICE methods, although it is adapted here to include the cost constraint. The idea of constraining an upper bound on the cost rather than the estimated cost itself is also well-established. Ultimately though, I think the extension of an existing class of methods to a new domain they have not been applied in yet is sufficient novelty for publication.



**Summary Of The Paper:**

The authors propose a DICE-family method for solving constrained offline reinforcement learning problems. To do this, they adapt ideas from OptiDICE and find a reduction from a nested constrained optimization problem to a single unconstrained optimization problem that can be efficiently represented with a neural network. Additionally, they draw on ideas from CoinDICE to estimate a confidence interval over the cost, which makes their method better at obeying constraints. The  authors compare their method to a number of baselines on both random grid worlds and continuous environments, and find that COptiDICE achieves both good performance and better constraint satisfaction than alternative methods.

**Summary Of The Review:**

The paper is well-founded and feels like a "correct" solution to this problem, and the empirical results show the method performs well in practice. I

---

> ### Author Response · Authors · 2021-11-22
> **Response to Reviewer NsmH**
>
> Thank you very much for your feedback.
>
> Regarding the novelty of our work, we would like to highlight that we provide an efficient and principled method to address the challenge in offline **constrained** RL. Specifically, to mitigate the challenge of the constraint violation, our COptiDICE constrain the **upper bound** of the cost value, where the cost upper bound is estimated efficiently by solving a **single minimization problem** via reusing the stationary distribution corrections obtained by DICE-based RL. This is in contrast to CoinDICE, where it requires solving a *nested max-min problem* for estimating the cost upper bound (see Eq 18 vs. Eq 20). In conclusion, we would like to highlight that our COptiDICE is not a trivial extension of both OptiDICE and CoinDICE.

---

> > ### Comment · Reviewer_NsmH · 2021-12-06
> > **Thank you for clarifying**
> >
> > Thank you for clarifying the difference with CoinDICE. I will raise my score accordingly

---

### Official Review · Reviewer_QeFP · 2021-11-14

**Correctness:** 4
**Technical Novelty And Significance:** 3
**Empirical Novelty And Significance:** 2
**Recommendation:** 6
**Confidence:** 3

**Main Review:**

Strengths:
1) they propose to optimize the state-action stationary distribution directly, which avoids the instability caused by triple optimization problems for the actor, the critic and the cost Lagrange multiplier with three different objective functions in the existing work that manipulates both Q-function and policy.


Weakness:
1) important baselines (state-of-the-art offline RL with Lagrangian approach) are missing:
- in CMDPs: besides MLE CMDP, only the variant of SPIBB, an offline RL method for tabular MDPs, is compared with the proposed method (applicable to continuous tasks). Behavior cloning is also missing in this setting to see whether the proposed method is just remembering the collected safe dataset.
- in continuous control tasks: CRR, sota unconstrained offline RL without any modification to handle constraints, is compared with the proposed method
2) It is always difficult to reach a confident conclusion that one method is better than others when a trade-off between rewards maximization and constraint satisfaction is needed. In both tabular CMDPs and continuous control tasks, although the proposed method shows better performance in satisfying the constraint, the constraint cost is always very close to the predefined single cost threshold. Ablation study on constraint cost threshold is required to both show the sensitivity of the proposed method to different cost threshold and consistent advantage over other baselines.


Suggestions
1) In tabular CMDPs, both constraint-satisfying policy and constraint-violating policy are used to collect datasets. In continuous control tasks, the online constrained RL agents are used to collect data. However, in the real world, agents normally act safely but have some unsafe attempts. It would be more interesting if a mixture dataset collected by both constraint-satisfying and constraint-violating policy is utilized for constrained offline policy learning.
2) There is a concurrent work, constraints Penalized Q-Learning (CPQ) [1], similarly aiming at offline constrained RL. It would be great if some comparisons are studied between the proposed method and this work (either methodology aspect or empirical aspect).
3) It would be great if there is some exploration describing how the proposed work can be adapted to hard-constraints scenarios since safety constraints in the real world are often hard constraints as well as constraints imposed by physical laws.

[1] Xu, Haoran, Xianyuan Zhan, and Xiangyu Zhu. "Constraints penalized q-learning for safe offline reinforcement learning." arXiv preprint arXiv:2107.09003 (2021).

**Summary Of The Paper:**

This paper has presented a DICE-based offline constrained RL algorithm for constrained RL. Experimental results on tabular CMDPs and continuous control tasks show that the proposed method can achieve a better trade-off between reward maximization and constraint satisfaction.

1st contribution: They firstly proposed to tackle constrained offline RL by solving a single minimization problem.

2nd contribution: To mitigate constraint violation in practice, they exploit the distribution correction obtained by solving the RL problem for cost upper bound estimation and then constrain the upper bound.

**Summary Of The Review:**

The paper proposes to optimize the state-action stationary distribution directly without considering the unstable triple intertwined optimization porblems in actor-critic-based constrained RL algorithms. However, the effectiveness of the proposed method in obtaining a trade-off between reward maximization and constraint satisfaction will be more convincing if 1) more straightforward adaptation of state-of-the-art offline policy learning algorithms are considered as baeslines; 2) more ablation study to analyze the sensitivity of the performance against different cost thresholds.

---

> ### Author Response · Authors · 2021-11-22
> **Response to Reviewer QeFP**
>
> We thank the reviewer for the thoughtful feedback.
>
> **[Missing baselines]**
>
> We added the baseline algorithms suggested by the reviewer in the experiment.
> * For tabular CMDPs (Figure 1), we added the result for behavior cloning (BC), where its performance becomes identical to the data-collection policy as the size of the dataset increases. It confirms that our method is not just remembering the dataset.
> * For continuous control tasks (Figure 2-3), we implemented C-CRR (CRR with Lagrangian relaxation to deal with constraints), where a cost critic and a cost Lagrange multiplier are additionally introduced to CRR. C-CRR jointly optimizes actor, reward critic, cost critic, and Lagrange multiplier with four different objective functions. As can be seen from Figure 2-3, our COptiDICE still outperforms C-CRR, showing a better trade-off between reward-maximization and constraint satisfaction.
>
>
> **[Ablation study on different cost thresholds]**
>
> We conducted additional experiments on different cost thresholds (Figure 4 in Appendix E), where we tested varying cost thresholds $\hat{c} \in \{0.08, 0.09, 0.10, 0.11, 0.12\}$ in tabular CMDPs. Our COptiDICE still shows consistent advantages over other baselines in terms of constraint satisfaction and reward maximization.
>
>
> **[Experiments using mixture dataset]**
>
> For continuous control tasks, we have already used a mixture dataset in the experiments. For Figure 2, the mixture of 80% cost-satisfying trajectories and 20% cost-violating trajectories is being used. In Figure 3, we have shown the result using various mixture ratios between cost-satisfying trajectories and cost-violating trajectories: (80%, 20%), (50%, 50%), (20%, 80%). Our COptiDICE demonstrates the most robust behavior to avoid constraint violation across diverse dataset mixture ratios.
>
> For tabular CMDPs, we conducted additional experiments to use a mixture dataset collected by both cost-satisfying policy and cost-violating policy, instead of using a dataset collected by a single policy (Figure 5 in Appendix F). Even when using the mixture dataset, our COptiDICE shows a stricter constraint satisfaction than baseline algorithms, showing a better trade-off between reward maximization and constraint satisfaction.
>
>
> **[Comparison with the concurrent work, CPQ]**
>
> The concurrent work, CPQ, also aims to solve offline constrained RL. CPQ shares the similarity with COptiDICE in that it also exploits *overestimated* cost value to deal with constraint satisfaction in an offline RL setting. Still, CPQ differs from COptiDICE in that CPQ relies on an actor-critic algorithm, while COptiDICE relies on stationary distribution optimization. We have discussed this shortly in Section 5. Since there is no publicly available code for CPQ currently, please understand that it was hard to compare with them directly in the experiments.
>
>
> **[Hard constraints scenario]**
>
> Thanks for your suggestion. In this work, we aim to compute a policy that satisfies the constraints in expectation. Extending our method to hard constraints scenarios (i.e. no constraint violation in every trajectory return) remains as future work. One simple way to impose the hard constraint in some limited situations is to adopt an additional layer to our optimized policy network (e.g. OptLayer [1] or SafetyLayer[2]) to make the sampled actions be projected to the safe actions.
>
>
>
> [1] Pham et al., "OptLayer - Practical Constrained Optimization for Deep Reinforcement Learning in the Real World", 2018 (arxiv:1709.07643)
>
> [2] Dalal et al., "Safe Exploration in Continuous Action Spaces", 2018 (arxiv:1801.08757)

---

### Author Response · Authors · 2021-11-22
**General Response**

We thank all the reviewers for their constructive feedback and comments. Below we highlight the main contribution of our work and the additional experiments added to the revised paper. The modifications in the revised paper are colored in blue.

**[Contributions]**
* We provide an efficient and principled method to address the challenge in offline *constrained* RL by proposing to constrain the upper bound of the cost.
* We firstly derived that constrained offline RL can be tackled by solving a single minimization problem by operating in the space of stationary distribution.
* We propose an efficient method to estimate the cost upper bound by solving a *single minimization problem* via reusing the stationary distribution corrections obtained by DICE-based RL. This is in contrast to CoinDICE, where it requires solving a *nested max-min problem* for estimating the cost upper bound.
* The proposed method significantly outperforms baselines in terms of reward maximization and constraint satisfaction.

**[Additional Experiments]**
* (Figure 1): For tabular CMDPs, we added the result for behavior cloning (BC).
* (Figure 2-3): For continuous control tasks, we added the result for C-CRR, the constrained variant of CRR with Lagrangian relaxation.
* (Figure 4 in Appendix E): We added an ablation study on different cost thresholds.
* (Figure 5 in Appendix F): We added tabular CMDP experiments with a mixture dataset.

---

### Decision · Program_Chairs · 2022-01-20

**Decision:**

Accept (Spotlight)

**Comment:**

This paper presents a new technique for constrained offline RL. The proposed method is based on reducing a nested constrained optimization problem to a single unconstrained optimization problem that can be efficiently represented with a neural network. The proposed algorithm is tested against several baselines on both random grid-worlds and continuous environments. Results clearly show that the proposed algorithm outperforms baselines while keep the provided constraints satisfied.

The reviewers agree that the paper is well-written, the proposed algorithm is novel and technically sound, and the empirical evaluation clearly supports the claims of the paper. There were some concerns regarding the novelty of this idea, but these concerns were properly addressed by the authors in the discussion.